# Process-based Estimate of Global-mean Sea-level Changes in the Common Era

Nidheesh Gangadharan[1], Hugues Goosse[1], David Parkes[2], Heiko Goelzer[3], Fabien Maussion[4], Ben Marzeion[5]

[1] Earth and Life Institute, Université catholique de Louvain, Louvain-la-Neuve, Belgium
[2] Department of Mathematics and Statistics, Lancaster University, Lancaster, United Kingdom
[3] NORCE Norwegian Research Centre, Bjerknes Centre for Climate Research, Bergen, Norway
[4] Department of Atmospheric and Cryospheric Sciences, University of Innsbruck, Innsbruck, Austria
[5] Institute of Geography and MARUM – Center for Marine Environmental Sciences, University of Bremen, Bremen, Germany

Correspondence to: Nidheesh Gangadharan (nidheeshag@gmail.com)

**Abstract.** Though the global-mean sea level (GMSL) rose over the twentieth century with a positive contribution from thermosteric and barystatic (ice sheets and glaciers) sources, driving processes of GMSL changes during the pre-industrial Common Era (PCE; 1 – 1850 CE) are largely unknown. Here, the contributions of glacier and ice sheet mass variations and ocean thermal expansion to GMSL in the Common Era (1 – 2000 CE) are estimated based on simulations with different physical models. Although the twentieth-century global-mean thermosteric sea level (GMTSL) is mainly associated with
temperature variations in the upper 700 meters (86% in reconstruction and 74±8% in model), GMTSL in the PCE is equally controlled by temperature changes below 700 meters. GMTSL does not vary more than ± 2 cm during the PCE. GMSL contributions from the Antarctic and Greenland ice sheets tend to cancel each other during the PCE owing to the differing response of the two ice sheets to atmospheric conditions. The uncertainties of sea-level contribution from land-ice mass variations are large, especially over the first millennium. Despite underestimating the twentieth-century model GMSL, there
is a general agreement between the model and proxy-based GMSL reconstructions in the CE. Although the uncertainties remain large over the first millennium, model simulations point to glaciers as the dominant source of GMSL changes during the PCE.

# 1 Introduction

Contemporary global-mean sea level (GMSL) rise is one of the key indicators of Earth's energy imbalance. For instance, the
GMSL rise (1.2 - 1.5 mm yr$^{-1}$) in the twentieth century (e.g. Hay et al., 2015; Frederikse et al., 2020) is linked to the fact that
nearly 90% of the excessive radiative heating of the climate system, due to greenhouse gas emission, has been stored into the
oceans (e.g. von Schuckmann et al., 2016; Church et al., 2013; Zanna et al., 2019; Meyssignac et al., 2019). The remaining
heat also has an impact on GMSL via altering the mass balance of continental ice (Antarctic and Greenland ice sheets and
glaciers) and changing the global hydrological cycle (Rignot et al., 2011; Shepherd et al., 2012; Church et al., 2013). In addition
to those responses to anthropogenic forcing, the internal variability in the coupled climate system is also suggested to explain
a part of the recent GMSL rise owing to the long memory of the oceans (Ocaña et al., 2016; Gebbie and Huybers, 2019).
Hence, the GMSL is integral to changes in the climate system in response to both forced and internal variability.

Recent studies have shown that more than 90% of the observed change in GMSL during the last few decades can be explained
solely by ocean thermal expansion and changes in the mass balance of continental ice storage (Leuliette and Miller, 2009;
Church et al., 2011, 2013; WCRP sea-level budget, 2018). Significant improvements in the understanding of changes in ocean
thermal structure as well as in the continental ice and water storage (e.g. Kjeldsen et al., 2015; Marzeion et al., 2015; Zanna et
al., 2019; Parkes & Marzeion 2018; Humphrey & Gudmundsson 2019) have indeed resulted in the closure of GMSL budget
for the entire twentieth century, pointing to the dominant role of those processes (e.g. Frederikse et al., 2020).

GMSL was about 120 meters below the current level at the last glacial maximum (about 21 ka BP). The subsequent deglaciation
caused the sea level to rise until the mid-Holocene (from ~16.5 ka BP to ~ 6 ka BP), followed by smaller amplitude variations
(Lambeck et al., 2014). The centennial-scale GMSL changes seen in proxy-based sea-level reconstructions over the pre-
industrial Common Era (PCE; 1–1850 CE) do not exceed 10-15 centimeters (Kopp et al., 2016; Kemp et al., 2018; Walker et
al., 2021). For instance, the proxy sea-level reconstructions indicate that the GMSL was in 600 CE ~ 5-10 cm above the 1850
CE level, and the multi-centennial rates do not exceed ± 0.2 mm yr$^{-1}$ during 1 – 1800 CE (Walker et al., 2021). The GMSL
rose rapidly by ~15 cm after 1850 CE to reach its current level. The rate of GMSL change during the twentieth century is thus
an order of magnitude higher than the multi-centennial rate during the PCE. In this context, the steady and rapid rise in GMSL
since 1850 CE is unprecedented over the last two millennia and largely a result of anthropogenically-driven surface warming
(Kopp et al., 2016; Slangen et al., 2016), which is identified to be unique in terms of rate and spatial coherency (Neukom et
al., 2019).

The driving processes of GMSL changes during the PCE are likely the same as those responsible for the recent sea-level rise
(e.g. Gregory et al., 2006; Ortega et al., 2013), but their relative contributions are largely unknown. In particular, considering
PCE as a period with weak anthropogenic perturbations allows one to examine the processes controlling GMSL from natural

forcing and internal climate variability. In addition to that, the response time of the climate components (ocean, glaciers and ice sheets) and the corresponding GMSL change can be of the order of several centuries (upper ocean and glaciers) or even millennia (deep oceans and ice sheets). Consequently, the understanding of current changes in GMSL calls for the analysis of past variability. For example, the cooling anomalies observed in the deep Pacific in the twentieth century were related to the

Little Ice Age (LIA) cooling and shown to offset the recent global heat gain in the upper ocean in some regions (Gebbie and Huybers 2019).

This paper focuses on the GMSL changes in the CE by analyzing the contributions of major components (ocean thermal expansion and changes in continental ice-mass balance) derived from model experiments. A comparison of model-derived

GMSL with proxy-based sea-level estimates (e.g. Kopp et al., 2016) is also provided. Such an exercise is important because estimates of different contributing processes to GMSL can provide insights into potential differences in the mechanisms controlling those GMSL changes between different periods in the CE. Also, as the climate in the PCE is less impacted by the anthropogenic forcing and largely a result of natural variability, the findings place the current anthropogenic warming and sea-level rise in a broader context. Using model simulations, we mainly ask whether the GMSL changes over the CE can be

explained within the uncertainty estimates and what the major sources of uncertainty are. Also, which processes determine the centennial-scale variability seen in the proxy-based GMSL reconstructions during the CE?

## 2 Data and methods

GMSL varies either due to changes in the ocean mass (*barystatic* changes; Gregory et al., 2019) which primarily reflects changes in continental ice mass and terrestrial water storage, or by changing the density of the seawater (*steric* changes). The

relative contribution of terrestrial water storage changes to twentieth-century GMSL change is not negligible but predominantly associated with water impoundment in dams (Frederikse et al., 2020). The contribution is likely much weaker during pre-industrial conditions, and considering the difficulties of estimating precisely the climate-driven terrestrial water storage changes in the past, we do not consider it in this study. Hence, our barystatic sea-level estimates are confined to mass balance changes of ice sheets (Antarctic and Greenland) and glaciers. The global-mean *halosteric* sea-level changes due to

freshwater input to the ocean can be neglected, as the volume changes due to changes in the ambient salinity would be compensated by the salinity change of the added freshwater (Lowe and Gregory 2006; Gregory et al., 2019). Steric contribution to GMSL is hence reduced to global-mean thermosteric sea level (GMTSL).

### 2.1 Thermosteric sea level

GMTSL is estimated using the 3-D field of ocean temperature and salinity from *last millennium* simulations performed in the

framework of the Paleoclimate (Coupled) Modelling Intercomparison Project (PMIP3/CMIP5; Taylor et al., 2012; Braconnot et al., 2011; 2012) and with the LOVECLIM (Goosse et al., 2010; Shi et al. 2021) and CESM1 (Otto-Bliesner et al., 2016)

models (List of models is given in Table S1). GMTSL is computed as the area-weighted mean of thermosteric levels over the entire ocean surface area (A):

$$GMTSL = \frac{1}{A} \iint_{z=-H}^{0} \alpha \, \Delta\theta \, \Delta z \, dA$$

The thermal expansion coefficient ($\alpha$) at each standard depth level is estimated as:

$$\alpha^\theta = \frac{1}{\vartheta} \frac{\partial\vartheta}{\partial\theta}\Big|^{s,p} \, ,$$

where $\vartheta, \theta, s,$ and $p$, are specific volume, potential temperature, absolute salinity and pressure, respectively.

Computations covered 1 – 2000 CE for LOVECLIM and 850 – 2005 CE for PMIP3/CMIP5 models and CESM1. We used the
equation of sea-water state of Jackett and McDougall (1995) for the thermosteric sea-level computations rather than the recent equation (International thermodynamic equation of seawater – 2010), as most of the climate models considered in this study employ the former equation in their formulation. For GMTSL computations using simulations from the PMIP3 *last millennium* experiment (which covers 850 – 1850 CE), we considered mean ocean temperature for the period 1841-1850 CE as the reference field, and for CMIP5 *historical* experiments (1851 – 2005 CE), the chosen reference period is 1851-1860 CE. This
selection of consecutive reference periods allows the thermosteric levels from the two experiments, which are discontinuous for some models forbidding the definition of a common reference field for the two runs, to converge over 1841–1860 CE. Thus, all the GMTSL estimates presented in this paper (including LOVECLIM and CESM1) are referenced to 1841-1860 CE mean temperature field, and the computation is performed on the original ocean grid for each model (given in Table S1). Considering the deep ocean temperature drift and unavailability of control runs to correct it for the PMIP *last millennium*
experiments, we restricted the GMTSL computations to the top 700 meters for the PMIP3/CMIP5 models. However, full-depth is considered for LOVECLIM (there is no such climate drift as a long spin-up is performed before starting the simulation in 1 CE) and CESM1 (corresponding control runs available). Since the *GISS-E2-R* simulation performed in PMIP3 showed an apparent drift during the first 150 years (850 – 1000 CE), even for the top 700m, we have discarded that period for this particular model from ensemble computations.

**2.2 Antarctic and Greenland ice sheets**

The contribution from ice sheet mass variations (Antarctic and Greenland) is estimated from the finite-difference ice sheet model *IMAUICE* (de Boer et al., 2014) that has recently contributed to several inter-comparison exercises for future projections of the Greenland ice sheet (Goelzer et al., 2018; 2020) and the Antarctic ice sheet (Seroussi et al., 2019; 2020, Sun et al., 2020; Levermann et al., 2020). For the present application and due to the absence of high-resolution forcing records over the CE, we
force the models (configured for Greenland and Antarctica) with surface mass balance (SMB) calculated from a positive-degree-day model (PDDM, Huybrechts and De Wolde 1999) instead of prescribing SMB anomalies. The PDDM is driven by

spatially homogenous temperature anomalies relative to a 1960-1989 reference climate obtained either from climate model output or ice core-based temperature reconstructions.

### 2.2.1 Greenland

The model for Greenland is set up at 16 km horizontal resolution and uses the shallow ice approximation. Floating ice is not considered and is removed when it occurs. In the absence of i) useful constraints on marine-terminating outlet glacier evolution over the CE, ii) applicable forcing and iii) sufficient process understanding, the model does not consider explicit ice-ocean interactions and is driven by SMB forcing alone. The model initialization builds on earlier work and starts from an existing thermodynamically coupled steady state with constant, present-day boundary conditions (IMAUICE1 in Goelzer et al., 2020).

From here, the model is relaxed for 10 kyr with a fixed ice temperature to the PDDM SMB forcing with a surface temperature anomaly of zero degrees to produce a nominal initial state for CE simulations.

For simulations forced using PMIP3 GCM outputs, the initial state is assigned to the year 850 CE, and the model is run forward with GCM-derived spatially constant 2-m air temperature anomalies as input to the PDDM. Simulations using output from

LOVECLIM are set up similarly but cover the entire CE, with the initial state assigned to CE 1. The same is true for forcing derived from the last millennium reanalysis data version 2 (LMR; Tardif et al., 2019). The GCM-forced experiments are complemented by simulations driven with spatially constant temperature anomalies derived from an ice core record (Kobashi et al., 2011), spanning 1-2000 CE. We have introduced two types of perturbations to test uncertainty in the initial state. In the first case, the initial state is perturbed for 500 years before the start of the simulations with a constant temperature offset

between -0.5 K and +1.0 K. In the second set of experiments, the model is run to new steady states with temperature offsets between +0.4 K and +0.6 K. The design of these additional experiments implies that the forcing over the CE is identical for the ensemble of simulations, while the model responds differently to the initial perturbations. To avoid double-counting, our simulations do not account for peripheral glaciers weakly connected to the Greenland ice sheet, which are then included in our glacier model (section 2.3).

**2.2.2 Antarctic**

The model for Antarctic simulations is run at a 32 km horizontal resolution and uses a combination of shallow ice and shallow shelf approximations, with velocities added over grounded ice to model basal sliding (Bueler and Brown, 2009). We use the Schoof flux boundary condition (Schoof, 2007) at the grounding line with a heuristic rule, following Pollard and DeConto (2012). Model initialization again builds on an existing present-day ice sheet steady state (IMAUICE2 in Seroussi et al., 2020),

which is first relaxed with fixed ice temperature for 10 kyr to the PDDM SMB and zero sub-shelf basal melting. We then continue for another 5 kyr with background sub-shelf basal melt rates estimated for the modelled ice draft using the shelf melt parameterization of Lazeroms et al., (2018) with a thermal forcing based on the World Ocean Atlas (WOA; Garcia et al., 2019) at 400 m depth. Assuming a colder ocean for the first millennium CE, and since we couldn't find stable, steady-state grounding

line positions for the original thermal forcing, we introduced an offset of -0.5 degrees to the WOA thermal forcing. The resulting melt rates are largest in the Amundsen sea embayment and have a maximum of 25 m yr$^{-1}$ at the Pine Island glacier grounding line.

Sub-shelf basal melt rate anomalies for the transient GCM-forced experiments are derived using spatially uniform ocean temperature anomalies averaged over 400-600 meters from models in combination with a high sub-shelf melt sensitivity of 11 m a$^{-1}$ K$^{-1}$ (Jenkins, 1991; Payne et al., 2007; Levermann et al., 2020). An example of this ocean temperature extraction is shown for LOVECLIM (Figure S1). The ice sheet model is also forced with model air temperature anomalies driving the PDDM SMB calculations. Similar to the Greenland simulation, the experiments cover the period from 850 CE onwards, except for LOVECLIM- and LMR-based runs, which cover the entire CE. The GCM-based experiments are complemented by simulations forced with a spatially constant temperature anomaly from a reconstruction based on ice-core records (Stenni et al., 2017), spanning the entire CE. In the absence of a usable proxy record for sub-shelf ocean temperatures, the background sub-shelf basal melt rates are held constant in these experiments. As Greenland's second set of initial state perturbations described, we produce two alternative initial steady states using an air temperature offset of ±0.5 K.

## 2.3 Glaciers

The glacier volume change estimates are made using the *Open Global Glacier Model (OGGM,* Maussion et al., 2019) version 1.4 (Maussion et al., 2021). OGGM is an open-source model which couples a surface mass balance model with a model of glacier dynamics. OGGM is used to model the annual rate of glacier mass change for 18 of the 19 glaciated regions defined in the Randolph Glacier Inventory (RGI; Pfeffer et al., 2014), with the Antarctic/sub-Antarctic region not modelled due to limitations of the baseline climatology dataset. We used gridded monthly temperature and total precipitation records from the last millennium reanalysis (LMR) data version 2 (Tardif et al., 2019) to drive the model. OGGM determines the temperature and precipitation at each glacier location by applying these as anomalies to a reference climate. We have not used PMIP climate model results because of the potential biases in those models that would require specific corrections before driving OGGM adequately, and deriving those corrections is out of the scope of the present study (Parkes and Goosse, 2020).

Runs with two different reference climatic conditions are performed: one using CRU TS 4.01 (Climatic Research Unit gridded Time Series 4.01; Harris et al., 2020) mean climate from 1951-1980 (glacier simulation covers the period 1 – 2000 CE), and the other using ERA5 (covers 850 – 2000 CE), which is a recent update of the ERA-interim data as documented in Hersbach et al., (2020). Temperature and precipitation at the reference grid elevation for each of the two datasets are scaled to the glacier surface at each OGGM grid point using a default temperature lapse rate of -6.5°C / km between the reference elevation and the glacier surface elevation and a uniform precipitation multiplier of 2.5 (CRU) and 1.6 (ERA5) to account for enhanced precipitation and lateral transport of snow by wind and avalanches in mountainous topography. The model is calibrated to *in-*

*situ* observations provided by the World Glacier Monitoring Service (WGMS, 2020) and then corrected to match regional mass-change estimates by Hugonnet et al. (2021).

The contributions of positive degree-months for ablation and solid precipitation for accumulation are combined to calculate mass balance, which is used to update glacier geometry annually. In this study, frontal ice ablation of tidewater glaciers is not simulated explicitly. The initial state of mountain glaciers at the beginning of the millennial simulations is unknown: we therefore use the year ~ 2000 state (the area from RGI and volume from Farinotti et al., 2019) as initial conditions. The first decades (most glaciers) to centuries (large, flat ice fields) of the simulations are therefore more uncertain and can be considered as a "spin-up". More details of OGGM workflow can be found in Maussion et al. (2019), and further background on the mass balance calculation is available in the precursor to OGGM described in Marzeion et al. (2012).

The simulated volume ($V_{tot}$) for each region is corrected to remove the below-sea-level component ($V_{bsl}$), using a fixed proportion by region from Farinotti et al. (2019), and the Sea-level Equivalent (SLE) of the final volume (contribution to GMSL) is calculated as:

$$SLE = \frac{V_{tot} - V_{bsl}}{A_{ocean}} \frac{\rho_{ice}}{\rho_{water}} \quad ,$$

assuming a bulk ice density ($\rho_{ice}$) = 900 kg m$^{-3}$, ocean area ($A_{ocean}$) = $3.625 \times 10^8$ km$^2$ and density of freshwater ($\rho_{water}$) = 1,000 kg m$^{-3}$.

## 2.4 Sea-level reconstructions

GMSL derived from proxy-based sea-level reconstruction for the CE from Kopp et al. (2016), Kemp et al. (2018) and Walker et al. (2022) are considered for comparison with our model GMSL. Those GMSL reconstructions are iterations of a spatio-temporal statistical model applied to a growing database of the CE proxy reconstructions. In this spatio-temporal model framework, GMSL is an estimate from the signal "common" to all sea-level records in the CE proxy database. Since the GMSL is the "globally uniform" term among sites in the spatio-temporal model, the method could give a true estimate of GMSL in the presence of spatially complete data. Consequently, the quality of the estimate depends on the geographic distribution of proxy records, which is very uneven (however, some sensitivity tests to explore the effect of the geographic distribution of proxy records have been done in Kopp et al. 2016). As the Walker et al. (2022) reconstruction is based on the latest update of the proxy sea-level database, and the Kemp et al. (2018) and Kopp et al. (2016) curves do not differ much over the CE, we show GMSL from Walker et al. (2022) and Kemp et al. (2018) in our model comparison. Also, in Kemp et al. (2018), the GMSL during -100 – 100 CE is made equal to GMSL over 1600 – 1800 CE to avoid a spurious global sea-level trend component originating from regional changes. However, such a constraint is not employed in the Walker et al. (2022) reconstruction. As a result, there is an apparent difference between the GMSL curves in these two reconstructions before ~

600 CE (c.f. Fig. 4). Note that the updated database of Walker et al. (2022) would also have an impact on the difference between the two curves shown in Fig. 4.

We also compare our model GMTSL with the reconstructed GMTSL estimates from Zanna et al. (2019) over 1870 – 2018. Since Zanna et al. (2019) already compared their reconstruction to different observation-based oceanic heat content estimates (e.g. Levitus et al. 2012; Ishii et al. 2017), we do not show all those available products in this paper for the twentieth-century comparison. Reconstructions of ocean temperatures over the CE are limited to either sea surface temperature derived from paleoceanography (proxy) data (e.g. PAGES Ocean2k Synthesis Data; McGregor et al., 2015) or spatially averaged oceanic

heat content estimates generated through inverse modelling and using available instrumental and paleo-data (Gebbie and Huybers 2019). Though such datasets have helped to understand certain key features of ocean climate variability during the CE, they do not provide a direct estimate of the contribution of ocean changes to GMSL. Hence, we do not attempt to compare our model thermosteric variability with any of those datasets in this paper. Also, as the GMSL reconstruction from Walker et al. (2022) and Kemp et al. (2018) already incorporated the tide-gauge-based twentieth-century GMSL (e.g. Hay et al., 2015),

we do not show those available twentieth-century GMSL reconstructions in this paper.

## 2.5 Uncertainty estimates

As described in the previous sections, our model experiments span two distinct periods (either 1 – 2000 CE or 850 – 2000 CE) depending on the input fields used to run the ice-sheet model or the reference climate used in our glacier model. Our thermosteric estimates also cover these two periods depending on the model (1 – 2000 CE for LOVECLIM and 850 – 2000

for the rest of the models). Hence, we present our model-derived sea-level components and the final GMSL estimates as two groups, namely *EXP-I* (simulations covering 1 – 2000 CE) and *EXP-II* (850 – 2000 CE)*,* primarily based on the period of model simulations. Table 1 summarises the input/reference fields used in these two groups. The two groups of simulations allow us to test the sensitivity of our model runs to different input fields and initial climate states. In addition to that, uncertainty derived from a "single-model large ensembles" (like the one performed with LOVECLIM GMTSL) provides an opportunity

to isolate uncertainty arising solely from internal climate variability, while the uncertainty from the PMIP models (or PMIP-based ice-sheet simulations; *EXP-II*) additionally represents differences in model physics. Also, the simulations directly driven by reconstructions, like the ice-core-based temperature estimates, provide alternative estimates not influenced by climate model potential biases.

### 2.5.1 Glacier uncertainty

To estimate model uncertainty for the glacier contribution, we combined the impacts of intra-regional and inter-regional uncertainty. It should be noted that, since all samples (regional glacier volume simulations) are taken from a *single* set of OGGM runs with a fixed set of parameters, this does not represent uncertainty in the model setup. It only represents uncertainty based on a) varying confidence in glacier inventory completeness or representativeness and b) OGGM's ability to effectively

model ice masses by varying glacier-by-glacier and region-by-region using the applied forcing. Intra-regional uncertainty is estimated with a 'leave X out' method by creating a set of reconstructions of volume for each region using a random sample of 50% of glaciers in the region and scaling the volume time series for the sample to match the total regional volume in 2001 as taken from Farinotti et al. (2019). A spread of 100 independently sampled volume time series is used to determine a time series for regional standard deviation, which is incorporated into the compound uncertainty estimates as described below.

Inter-regional uncertainty is also estimated with a 'leave X out' method by creating a set of global volume reconstructions, each leaving out three top-level RGI regions. The contribution of each region is then perturbed according to the regional standard deviation calculated as above. For each region in the sample of regions, a single value is sampled from a normal distribution with a mean of 0 and a standard deviation of 1. Then the standard deviation time series for the region is multiplied by that single value and added to the regional time series. Perturbing regional time series in this way results in a more realistic range than simply adding (a likely underestimate) or normalized multiplying (a likely overestimate) the independent uncertainty ranges from the intra-regional and inter-regional samples. The sample of perturbed regional time series (with the three top-level regions removed) is then added together and scaled to match the total (including all modelled RGI regions) global volume in 2001. We did this for 1000 independently sampled leave-3-out sets of regions and formed the confidence interval for combined intra- and inter-regional uncertainty as to the 1-standard deviation.

### 2.5.2 Uncertainty in rest of the processes

As shown in Table 1, our independent estimates of thermosteric and ice-sheet contributions are limited to less than ten cases and not consistent for these two processes. On the other hand, we have generated 1,000 synthetic curves and derived confidence levels for the glacier sea-level simulations, as described in the previous section. To have a consistent set of estimates for thermosteric and ice-sheet contribution, we employed a Monte Carlo method by generating 1,000 realizations of available estimates in each contributing process. Ensemble members are generated by randomly selecting and perturbing one of the available estimates at a time. Specifically, we perturbed the estimate by drawing random numbers (white noise) from a Gaussian distribution using the *a priori* standard deviation (which is taken as the RMSE between the ensemble mean and the randomly selected estimate) and adding those random numbers to the selected estimate. Note that this method does not include any other specific process that misses in our modelling experiments but acknowledges the remaining uncertainty (e.g. uncertainty arises from model initialization, different input data, or differences in model physics) and propagates the overall uncertainty to the final GMSL curve in a consistent way. An additional remark here is that, for thermosteric estimates from EXP-II, for which the computation is restricted to the top 700 meters owing to the deep-layer temperature drift, we added a *below-700-meter contribution* of 0.85 cm to the *a priori* standard deviation. This deep-layer contribution is estimated as the mean RMSE between the *full-depth* and *top-700m* GMTSL estimates from the LOVECLIM model over 850 – 2000 CE (see Fig. 1). As the sea-level estimate varies over much lower frequencies (as seen in Figure 2, for example), and the yearly white noise brings unrealistic high-frequency perturbations to the chosen sea-level estimate, we converted the generated white noise

($W$) to a red noise ($R$) using the lag-1 autocorrelation ($r$) of the chosen estimate before adding it to the chosen estimate, following Bretherton (2014):

$R_t = rR_{t-1} + (1 - r^2)^{1/2} W_t$

Since all the estimates are originally referenced to 1841 – 1860 CE, consistent with our thermosteric computations, we scaled the generated red noise with a scale factor that varies between (0, 1) according to the time-varying difference between the ensemble mean and the chosen estimate. This scaling ensures that we do not perturb the sea-level time series uniformly in time
but provide weightage for the chosen reference period. In other words, our uncertainty estimates are not *absolute* but relative to the reference period chosen, being smaller for the reference period by construction. The entire process is repeated to yield a thousand realizations of each contributing component and GMSL, for which all known sources of uncertainty and the spread among different estimates have been propagated. 1-standard deviation of these large ensembles is shown as the uncertainty of our central estimates. We compute the rate and budget of GMSL for each ensemble member and subsequently derive the mean
and confidence intervals from the large ensemble.

## 3 Results

### 3.1 Thermosteric sea level

Figure 1 shows the GMTSL computed over the entire depth (Fig. 1a), top 700 meters (Fig. 1b), and below 700 meters (Fig. 1c) from the LOVECLIM model for the CE. Our primary goal here with Figure 1 is to illustrate the relative contribution of
the *upper* (top 700 meters) and *lower* (below 700 meters) layer temperature variations to the total (computed over the entire depth) GMTSL changes over the last two millennia. As stated in section 2, nearly all the PMIP simulations exhibit a strong deep-layer temperature drift, and we confined our GMTSL computations to the upper layer for those models. Therefore, before describing the GMTSL variability from PMIP and other models (shown in Fig. 2a), we will examine the extent of deep-layer contribution using LOVECLIM and derive an uncertainty estimate for the PMIP GMTSL arising from deep-layer variability.
The LOVECLIM GMTSL estimates, separately for the two layers, show that the contribution of deep-layer variability may have an equal role in determining the total variability during the PCE, as described below.

Over the 1900 – 2000 CE, 86% and 74±8% of the total GMTSL rise is associated with the thermal expansion in the upper 700 m layer of the world oceans, as seen in Zanna et al., (2019) and LOVECLIM, respectively (Figure 1). These figures are
consistent with other studies that showed that the ocean heat content changes over the last fifty years are primarily contained in the upper layers of the world ocean (e.g. Levitus et al., 2012; Church et al., 2013). For instance, total GMTSL rose about 5-6 cm since 1900 CE, and the upper-layer GMTSL shows ~ 4 - 5 cm contribution over the same period, as seen in both Zanna

et al., (2019) and LOVECLIM (Fig. 1a, b). The prominent role of the upper layer in shaping the twentieth-century GMTSL change also indicates the absorption of anthropogenic heat in the upper oceanic layers.


However, the relative contribution of the oceanic upper and lower layers to the total GMTSL changes varies significantly over the PCE, as shown in Figure 1a. For instance, the upper layer cooling contributes only half of the total GMTSL decrease during 1500 – 1750 (LIA). An increase in the upper layer GMTSL during 1250 – 1500 was offset by a deep-layer cooling (Fig. 1b, c) and resulted in a weak total GMTSL rise over that period (Fig. 1a). The contribution of the upper layer to the GMTSL fall

during 250 – 500 (500 - 750) is about 56% (59%), suggesting that the cooling below 700 meters has an equally important role over those periods. The lag in the lower-layer thermosteric rise compared to the recent warming of the upper ocean (~ since 1850 CE; Fig. 1b, c) could be due to the extended deep-layer cooling from LIA, as shown in Gebbie and Huybers (2019). Similarly, a rise in the upper 700 m thermosteric sea level during 1250 – 1400 CE might be a rebound of the upper ocean from volcanic cooling induced by the strong 1257 Samalas eruption (see Sigl et al., 2015). However, the deep ocean has still cooled

during this period (Fig. 1c). In general, the differing thermosteric change in the upper and lower ocean over the CE indicates two distinct time scales of ocean response, the deep layer being much slower than the upper ocean. Note that the uncertainty of lower-layer GMTSL contribution (Fig. 1c) is comparatively larger than the uncertainty of upper-layer contribution throughout the last two millennia. The standard deviations of full-depth, upper-, and lower-layer GMTSL during 1 – 1850 CE (PCE) are 0.62±0.05 cm, 0.42±0.02 cm, 0.57±0.08 cm respectively. Those figures indicate that the deep-layer temperature

changes have an equally important role in shaping the total GMTSL variability in the PCE compared to temperature changes in the upper 700 m of the ocean.

The GMTSL estimates from all the available simulations show that the amplitude of GMTSL changes in the PCE is small compared to the twentieth-century rise (Fig. 1 & 2a). GMTSL does not vary more than ± 2 cm during the PCE (valid for both

upper and lower layers). In general, GMTSL from PMIP/CMIP (EXP-II) shows a similar evolution since 850 CE compared to the LOVECLIM, except for a larger uncertainty and a slight underestimation of the twentieth-century rise in CMIP5. The twentieth-century GMTSL rise is about 5 cm in reconstruction and LOVECLIM (Fig. 1a) but ~3 cm in CMIP models (Fig. 2a). Another notable feature is the short-term episodic falls (notably in 1259, 1453, 1601, 1641, 1809, 1815, and 1831) in GMTSL over the last millennium, evident in both LOVECLIM and PMIP simulations (Fig. 2a). These episodic falls in GMTSL

result from the reduction of the oceanic heat content from anomalous radiative forcing triggered by strong volcanic eruptions reported at those times (e.g. Sigl et al., 2015; Ortega et al., 2013; see fig. S2).

Figure 2a shows that the GMTSL increased during the first three centuries of the CE (so-called Roman Warm Period) and then declined to 700 CE, a period characterized by cold and dry climate conditions, referred to as the Late Antique Little Ice Age

(Helama et al., 2017). GMTSL then rose about 1 cm toward the Medieval Warm Period (~1200 CE) before declining again during the Little Ice Age (LIA). Those centennial-scale changes are also evident in the 250-year rate (*rate* here onwards) of

GMTSL (Fig. 3a), as it exhibits positive and negative values over centennial periods, following the "climate epochs" mentioned above. The rate curves are nearly identical for EXP-I and EXP-II over the CE, except over the last two centuries, where the rate is comparatively weak for EXP-II (this is expected because the twentieth-century rise is slightly underestimated in EXP-II, as shown in Fig. 2a). It is also interesting to note, in Figure 3a, that the global-mean surface temperature rate indeed agrees with the GMTSL rate over certain periods (by "agreement", we mean the sign of the rate over multi-centennial periods, for instance, during 0 – 600 CE, 1200 CE – present), but disagrees during 600 – 1200 CE. The GMTSL (global-mean surface temperature) rate is well within ±0.1 mm yr$^{-1}$ (±0.1 $^0$C century$^{-1}$) during the PCE, and the rate increases to ~ 0.15 mm yr$^{-1}$ (0.15 $^0$C century$^{-1}$) during the last two centuries (Fig. 3a).

## 3.2 Barystatic sea level

GMSL changes due to Antarctic mass balance variations over the instrumental period (Church et al., 2013; Frederikse et al., 2020) and future projections (Moore et al., 2013; Palmer et al., 2020; Seroussi et al., 2020) are highly uncertain. The uncertainties are also prominent over the PCE, with the uncertainty range (shading around the mean) of our estimate of the Antarctic ice sheet's contribution to sea-level changes (Fig. 2b) and its rate (Fig. 3b) including both positive and negative values for the majority of the period. The range of probabilities at the beginning of the first millennium in EXP-I (Fig. 2b) indicates either a sea-level fall or rise, depending on the initial state. The central estimate (ensemble mean) of the Antarctic sea-level contribution, however, shows a long-term fall over the first millennium (1 – 1000 CE) and a reversal in sign further until the early twentieth century (Fig. 2b). The positive sea-level contribution during the second half of the PCE (1000 – 1900 CE) is further supported by the PMIP-based simulations (EXP-II). However, the uncertainty is even larger in this case (Fig. 2b). EXP-II indicates a weak positive mass balance (negative sea-level contribution) during the twentieth century, while the twentieth-century change is nearly zero in EXP-I (Figure 2b).

The inferred uncertainties are also evident in the rate of sea-level change, as seen in Figure 3b. The central estimate of the rate (from EXP-I and EXP-II) over the entire period is in line with the sea-level or mass balance change described above. The rate is negative during 1 – 1000 CE and then becomes positive for the rest of the period (Fig. 3b). The twentieth-century decline in sea level is reflected in the sea-level rate, as the rate decreases towards the end of the period (the rate is still positive as our window of rate computation is 250 years). The surface temperature over Antarctica in the past two millennia (Stenni et al., 2017) exhibits an inverse relationship to sea level over multi-centennial periods (Fig. 2b). Our experimental design can explain this relationship as a warmer climate generally enhances precipitation over Antarctica and decreases the GMSL (Frieler et al., 2015; Medley and Thomas, 2019).

Greenland's contribution to GMSL exhibits substantial centennial-scale variability with a positive long-term trend (~ 4 cm rise) during 1 – 750 CE, probably in response to the large surface temperature variability over Greenland during this period (Figs. 2c & 3c). Despite considerable uncertainty, the contribution of Greenland ice-sheet mass changes to sea level was

probably well below the current level at the beginning of the CE (Fig. 2c). Figure 3c indeed shows that the sea-level variation during the 1 - 750 CE (rate varies between 0 – 0.2 mm yr$^{-1}$) has substantial centennial-scale changes and follows the surface temperature variations (varies between ±1.5 $^0$C century$^{-1}$) over the same period. The sea-level decline during 750 – 1850 CE (~ 2 cm) is also in line with the surface temperature fall in the same period in the two experiments. In general, the sea-level rate is positive (negative) during the first (second) millennium (Fig. 3c), opposite to those millennial-scale changes seen in

Antarctic contribution to sea level (Figs. 2b & 3b), suggesting a differing response of the two ice sheets to surface temperature changes. Greenland surface temperature and its sea-level contribution show an in-phase variability, as higher temperatures induce more melting of the Greenland ice sheet and a positive sea-level change. On the other hand, Antarctic sea-level contribution and its surface temperature have an 'inverse' relationship (Fig. 2b) as the temperature increase over Antarctica leads to increased mass accumulation and a decrease in sea level, as noted above. Thus, the dominance of different mass

balance processes explains the differing response of the two ice sheets over the CE. Though the recent warming over Greenland started as early as 1800 CE and temperatures rose by approximately 1 - 2 $^0$C to the reference period (1841-1860), no clear sea-level response was observed during this period (Figs. 2c & 3c).

Results from OGGM (Fig. 2d) suggest that the GMSL, as a response to global glacier mass balance changes, rose gradually

over 1-500 CE (with a sea-level equivalent of ~ 10 cm) and then exhibited a long-term fall until the early twentieth century (~ 6 cm). There is a positive sea-level contribution (~2 cm) during the subsequent decades (1920 – 2000). Note that the large glacier sea-level contribution over the first few centuries of the CE could be partly a model "spin-up' response (as highlighted by the red dashed line in Fig. 2d). Nevertheless, Figure 2d indicates that glaciers are the largest source of GMSL changes during the PCE, with an amplitude of associated sea-level variation much larger (2.8 cm standard deviation over the PCE) than

contributions from the rest of the sources (Fig. 2). However, the uncertainty is very large over the first millennium. As shown in Fig. 6 (which shows the sea-level contribution from each RGI region and corresponding surface temperature changes), there is considerable regional variability in the history of glaciers and surface temperature throughout the CE. The global sum of glacier contribution, shown in Fig. 2d, is thus a result of very different regional signals. The large millennium-scale change seen in the glacier sea-level contribution is also evident in the rate (Fig. 3d). The sea-level trend is positive during 1 – 800 CE

(rate gradually falls from ~ 0.3 mm yr$^{-1}$ at the beginning of the CE) and negative for the rest of the CE. The millennium-scale change is robust across the ensemble members, as inferred from the narrow uncertainty range enveloped around the ensemble-mean rate (Fig. 3d). As noted in section 2.5, the uncertainty estimate of the glacier contribution does not account for the full range (for instance, the single OGGM run restricts to integrate sensitivity of the model parameters and design into the uncertainty estimates), so the actual uncertainties might be larger than the one shown in this paper. The glacier-driven GMSL

changes are similar in both experiments (EXP-I and EXP-II) over their common period.

### 3.3 Combined estimate vs reconstruction

Figure 4a compares our model-based GMSL (i.e. the sum of the contributing processes shown in Fig. 2) with the GMSL from proxy-based reconstructions. An apparent difference between Kemp et al., (2018) and Walker et al., (2022) GMSL reconstructions at the beginning of the CE (green solid and dashed curves) is due to an imposed methodological constraint in the Kemp et al. (2018) reconstruction as noted in section 2.4. Model GMSL exhibits a broad agreement with proxy-based reconstructions, despite a few inconsistencies and large uncertainty in the first millennium. Our model GMSL indicates a steady rise of about 5 – 10 cm during the first five centuries of the CE. This rise is in line with Kemp et al., (2018; though the amplitude is smaller in reconstruction). However, the Walker et al. (2022) reconstruction shows a nearly steady GMSL over the same period (Fig. 4a). Reconstructed sea level shows a PCE maximum (~ 6 - 12 cm above the 1841 - 1860 CE level) during 500 - 700 CE, while the model GMSL shows a sea-level fall during this period. The PCE maximum in model GMSL (~ 8 cm) appears around 800 CE (Fig. 4a). Both proxy-based reconstructions and models indicate a long-term decrease in GMSL from 800 CE until the nineteenth century. There is, however, some multi-centennial variability within this long-term fall in reconstructions (for example, the GMSL fall during 1000 – 1250 and a subsequent rise during 1300 – 1500 CE), which is nearly absent in the modelled GMSL. Note that this multi-centennial variability is prominent in the Kemp et al. (2018) reconstruction but weaker in Walker et al. (2022), who used an expanded set of proxy data in their reconstructions (see section 2.4). Incorporating data from more locations could potentially dampen the regional sea-level signals in the global mean. Additionally, the model GMSL shows a ~5–8 cm GMSL rise in the twentieth century. This is only about half of what is seen in proxy-based reconstructions and reported elsewhere (e.g. Church et al., 2013; Hay et al., 2015). Our GMSL estimate from EXP-II shows similar evolution to that from EXP-I from 850 CE onwards (Fig. 4a).

Those salient features of GMSL evolution over the CE are also evident in Figure 4b, which shows a moving 250-year rate for GMSL (model and reconstructions) and global-mean surface temperature. The GMSL trend is positive during 1-500 CE in both model and reconstructions, and the rate varies between ~0.3 and 0 mm yr$^{-1}$. The model GMSL rate is below zero after 850 CE, ranging between -0.2 – 0 mm yr$^{-1}$. This negative rate corresponds to the long-term GMSL fall during the first part of the last millennium (Fig. 4a). The rate then becomes slightly positive by the end of the nineteenth century. The sea-level rate from reconstructions (either Walker et al., 2022 or Kemp et al., 2018) lies within the uncertainty of the model rate over 1 – 600 and 850 – 1800 CE. However, the rate is out of the model uncertainty range during 600 – 800 and post 1800 CE. The disagreement over 600 – 800 CE between the model and reconstruction results from inconsistent GMSL variability between the model and reconstruction during 500 – 700 CE, as seen in Figure 4a. Compared to reconstructions, the weak twentieth-century GMSL rise in the model is consistent with a too weak model GMSL rate (0.1 mm yr$^{-1}$ compared to 0.6 mm yr$^{-1}$ in reconstruction by the end of the nineteenth century; Fig. 4b).

The ensemble-mean thermosteric and barystatic (sum of ice-sheet and glacier contribution) sea level over the CE is separately shown in Figures 5a and 5b, respectively. In general, the thermosteric changes are much weaker than barystatic changes during the PCE (note that the scale is different for panels Fig. 5a & b). However, the twentieth-century model GMSL change is mainly attributed to thermosteric sea level rise. Also, while the barystatic changes occur mostly over millennial time scales, multi-decadal-to-centennial changes are evident in thermosteric variability (Fig. 5a). As noted in section 3.1, the multi-centennial GMTSL changes are linked to those regional climate epochs during the PCE. For example, the GMTSL was nearly down to the reference level (1841-1860 CE level) during 600 CE (Late Antique Little Ice Age – LALIA) and then rose by ~ 1 - 2 cm toward the Medieval Warm Period (MWP). GMTSL fell further during the Little Ice Age (LIA) and rebounded quickly during the Current Warm Period (CWP; post-1800). The respective contribution of different processes in shaping the GMSL (GMSL *budget*) during those climate epochs is shown in Figures 5c (1 – 600 CE), 5d (600 – 1200 CE), 5e (1200 – 1800 CE) and 5f (1800 – 2000 CE). Note that, as our GMSL estimates from EXP-II agrees well with EXP-I over their overlapping period (850 - 2000 CE), we show this analysis only for EXP-I.

Figures 5c-f illustrate that different processes contribute variable amounts (in terms of both rate and sign) to GMSL change in different periods, with glaciers as the dominant source throughout the PCE. Histograms in Figures 5c-f represent the ratio of the rate of individual contribution to the net GMSL rate (in percentage) over the selected period. For instance, the GMSL rise during the first 600 years and GMSL fall during 1200 – 1800 CE are driven mainly by glacier mass-balance changes (Figs. 5c&5e). Fig. 5d also shows that the weak GMSL change during 600 – 1200 CE (Fig. 4a) results from opposing contributions from its components. While the thermosteric and Greenland ice sheet exhibit a positive contribution (i.e. GMSL rise), the GMSL associated with changes in glaciers and Antarctic ice sheet shows a negative contribution (Fig. 5d) and resulting in a net GMSL change which is nearly zero over this period (600 – 1200; Fig. 4a). Note that the uncertainties are large for all the components except the glacier contribution during this period. All the GMSL components except the Antarctic ice sheet have a positive contribution to the net GMSL fall during 1200-1800 CE. As shown in Figure 5a, the model GMSL rise since 1800 CE is mainly linked to thermosteric rise with a weak contribution from barystatic sea-level components (Fig. 5b&f). Figures 5c-f indicate that the changes in the GMSL centennial rate (Fig. 4b) could be because the respective contributions to GMSL vary over such time scales.

## 4 Discussion

### 4.1 GMSL in the last two centuries

From instrumental records and models, it is virtually certain that the GMSL rose during the twentieth century with a mean rate of ~ 1.2 - 1.5 mm yr$^{-1}$ (e.g. Hay et al., 2015; Frederikse et al., 2020). The barystatic rise (about 1 mm yr$^{-1}$, including the terrestrial water storage contribution, which is ~ -0.21 mm yr$^{-1}$) is about twice the thermosteric contribution (~0.52 mm yr$^{-1}$) during 1900 – 2018 CE (Frederikse et al., 2020; Zanna et al., 2019). Though our model-based GMTSL estimates over the

twentieth century are consistent with those observation-based estimates (Fig. 1), there is an apparent underestimation of twentieth-century barystatic changes in our model simulations. For instance, a recent GMSL closure analysis by Frederikse et al., (2020) showed that glaciers are the largest source of the twentieth-century GMSL rise (contributing about 70% of the net barystatic rise and ~ 46% of the GMSL rise). Similar rates of glacier mass loss over the last century are also reported in earlier studies (e.g. Leclercq et al., 2011; Marzeion et al., 2015; Malles and Marzeion, 2021).

There are some notable differences between the formulation of our model-based estimates covering the entire CE and other estimates (some of which are mentioned above) focused on the twentieth-century barystatic GMSL rise. For example, Frederikse et al., (2020) accounted for the GMSL contribution from missing and disappeared glaciers (Parkes & Marzeion, 2018) and assumed a constant positive rate of Antarctic mass loss ($0.05 \pm 0.04$ mm yr$^{-1}$) before the satellite era. Such inputs are absent in our model simulations. Additionally, our glacier model (OGGM) has been initialized using the twentieth-century global glacier geometry with climate variables extracted from LMR. Assuming that the glacier geometry at the beginning of the CE is similar to the recent geometry may not be optimal. Still, we have no estimate at that time, and the drift from those initial conditions may affect the simulated twentieth-century glacier volumes. The prescribed initial glacier geometry (i.e. glacier state ~ 2000 CE) may not only impact the twentieth-century change but also impacts the glacier evolution during the entire CE. This is particularly an issue owing to the sizeable long-term glacier mass loss during the first millennium, as seen in Figure 2d. Towards the end of the simulation, the model state integrates biases over the whole period, particularly due to model drift and uncertainties in the forcing. We could 'correct' the state in 1800 to have better results over the last two centuries; however, the glacier distribution around ~1800 to initialize the model is not well-known, and the new model drift this 'correction' would induce at the start of the simulation is hard to estimate. Significant uncertainties also remain about the configuration and mass trend of both ice sheets at the onset of the PCE and their subsequent climate forcing and evolution. We have characterized this uncertainty using a wide range of initial conditions and climate forcing options in our modelling (section 2.2). The resulting ice-sheet simulations over the PCE represent an attempt to provide physically based ice sheet changes, complementing the other sea-level components. Further work should focus on better constraining the climate forcing specifically important for ice-sheet changes and developing paleo data that can inform about ice sheet evolution over the PCE. Despite those limitations, our model-based estimates provide insights into the GMSL changes in the PCE, as discussed below.

## 4.2 GMSL changes in the PCE

The GMTSL rise as a response to industrial climate warming started in the mid-nineteenth century in LOVECLIM (Fig. 1, 2a), following the global-mean surface temperature curve, but there is a lag of nearly half of a century in the PMIP ensemble mean (Fig. 2a) and probably in reconstruction (Zanna et al., 2019; Fig. 1). This lag could be one of the reasons for a relatively weak twentieth-century GMTSL rise in PMIP models. The relatively weak ocean thermal expansion in PMIP could have a link with the strong volcanic eruption of Krakatoa in 1883, as reported before in Gleckler et al., (2006), and possibly those eruptions earlier in the nineteenth century (See S2). LOVECLIM is able to capture those episodic GMTSL falls in response to

strong volcanic eruptions over the last millennium. However, the impact of the 19th-century volcanos seems weaker than some earlier ones in LOVECLIM (Fig. 1). LOVECLIM is a model of intermediate complexity, and the ocean's response to
volcanically-induced aerosol cooling in the background of anthropogenically-induced warming might be more complex. A correct ocean thermal response representation would strongly depend on model physics and experimental design.

Though the GMSL rose over the twentieth century with positive contributions from major sources, as shown in this study (except for a weak negative sea-level contribution from Antarctica in EXP-II) and elsewhere, the individual contributions to
GMSL in the PCE varied in sign and magnitude depending on the period considered (Fig. 5). Barystatic sea level dominates the GMSL variations throughout the PCE, with the largest (least) contribution from glaciers (Antarctica). This result is, in fact, identical to the relative contributions over the twentieth century (e.g. Frederikse et al., 2020). The amplitude of sea-level change due to glacier mass balance changes in the PCE (2.8±0.3 cm standard deviation; Fig. 2d) is remarkable. Despite considerable uncertainty, the glacier change is more prominent during the first millennium, which probably indicates relatively high glacier
surface mass balance sensitivity to initial glacier size and surface temperature changes.

The glacier contributions to sea-level change are broadly associated with the glacier-area weighted global-mean surface temperature evolution, as seen in Figure 2. For example, the surface temperature cooling during 1000 – 1800 CE is associated with a worldwide net glacier advance and a corresponding GMSL decrease (Figs. 2d & 3d). Note that the glacier advance and
surface temperature cooling during the second millennium are not globally uniform as there is considerable regional variability in the history of both glaciers and surface temperature throughout the PCE (Fig. 6). This is consistent with Neukom et al. (2019) finding that, unlike the twentieth-century global surface temperature rise, temperature variability during the pre-industrial period is not spatially uniform. Linking surface temperature more precisely with regional glacier changes would be difficult without further diagnoses. Nevertheless, we suggest that the changes in surface temperature and glaciers might occur
over distinct time scales. For instance, while the surface temperature over glacier regions shows strong decadal to multi-decadal variability, the large-scale glacier changes in the CE are mostly a centennial to multi-centennial response, for which the spatial coherency might appear relatively higher (Fig. 6 and Fig. 2d). As evident from Figure 6, not all glacier regions contributed equally to the GMSL changes in the CE, but a few areas (e.g. Greenland periphery, Russian Arctic) dominate the others (e.g. North Asia, Low latitudes). The distribution of glacier sea-level contribution in the CE seems to relate to the glacier initial
volume distribution (which is the twentieth-century glacier volume distribution as given in Farinotti et al., 2019). Going further on regional changes and the link between temperature and glacier changes is out of the scope of this present paper, which is focused on globally averaged signals (similarly, we restrain from describing the regional contribution of thermal expansion in different oceanic basins). A similar association between GMSL and regional surface temperature is also evident for Greenland and Antarctica (Fig. 2b, c & Fig. 3b, c). In the context of semi-empirical sea-level models (e.g. Oerlemans, 1989; Grinsted et
al., 2010; Jevrejeva et al., 2009; Kemp et al., 2011), those millennium-scale GMSL components presented in this paper,

combined with regional and global surface temperature, may potentially be helpful to resolve semi-empirical constants and response periods in a better way and can lead to useful hindcasts and projections.

On the other hand, the thermosteric sea level varies not more than ±2 cm during the entire PCE (0.62±0.05 cm standard deviation; Fig. 1 & 2a). Then, the GMTSL rose by ~ 4 - 6 cm during 1850 – 2000 CE, an unprecedented ocean heat content increase over the last two millennia. A weak GMTSL variability over the PCE seen in our experiments also provides an internal consistency, i.e. it supports our finding that barystatic changes contribute a large part of the GMSL variability in the PCE. Considering PCE as a period free from major anthropogenic emission and the corresponding weak GMTSL (and global-mean surface temperature) variability in this period (compared to post-1850 warming) suggests that centennial-scale ocean heat-content changes during the PCE, as a response to natural climate variability, are small. It also further supports the notions that recent changes are exceptional in the context of the past centuries and that the oceans have absorbed over 90% of the anthropogenic heat during the current climate warming (e.g. Church et al., 2011b).

The model and proxy-based reconstruction show some centennial-scale GMSL variability in the PCE (Fig. 4a). For example, GMSL varies up to 5 - 10 cm during 1 – 500 CE or 1300 – 1800 CE, and those figures are nearly half of the observed GMSL change over the twentieth century. The GMSL centennial changes over the PCE could primarily result from the slow and integrated response of sea-level components to surface perturbations and reflect the long-term persistence of oceanic thermal field and long response periods of barystatic components. Our results also suggest that some of those centennial-scale changes are comparable to the twentieth-century GMSL rise, for example, the sea-level change associated with Greenland variability during 1 – 500 CE (Fig. 2c). Those centennial-scale changes during the PCE indicate that the twentieth-century GMSL rise may also include a response to such natural variations. The offset of recent anthropogenic ocean warming by deep-layer cooling originated from LIA in the Pacific, as reported by Gebbie and Huybers (2019), is an example. We suggest that a similar influence of past variability can also be expected for barystatic sea level owing to its long response time scales, so that the recent GMSL change might be linked to variability in the past. Climate forcing integration can manifest as a lower-frequency change in the ocean, which can partly be misinterpreted as trends associated with deterministic forcing, as reported earlier in Ocaña et al. (2016). However, with the current simulations and analyses, it is hard to make firm conclusions on these aspects. Resolving the response time scales empirically and dedicated sensitivity experiments can provide more insights.

**5 Conclusions**

Although some earlier studies have discussed GMSL changes, either based on proxy-based sea-level reconstructions or semi-empirical methods (Kemp et al., 2011, 2018; Kopp et al., 2016; Grinstead et al., 2010; Walker et al., 2021), no attempt has been made to describe it using process-based modelling over the entire Common Era. This paper estimates the GMSL changes over the Common Era (1 – 2000 CE) by combining contributions from land ice (glaciers and ice sheets) mass variations and ocean thermal expansion simulated from different physical models. The GMSL contribution from different sources

(thermosteric and barystatic) varies considerably over periods during the PCE. Despite the large uncertainties, our model results suggest that the glacier contribution is higher than the contribution from other sources to GMSL changes in the CE. GMSL contributions from the Antarctic and Greenland ice sheets tend to cancel each other during the PCE owing to the differing response of the two ice sheets to atmospheric conditions. Thermosteric contribution to GMSL changes during the PCE does not reach more than ± 2 cm, and our results indicate that the ocean's upper 700 m and lower (below 700 m) layers

have an equal role in setting the global thermosteric changes during the PCE. Our results also suggest that the centennial-scale GMSL changes during the PCE, as seen in proxy-based reconstructions and models, would partly arise from the differing contributions from thermosteric and barystatic sources over centennial time scales (Fig. 5). There is considerable regional variability in the glacier contribution, and we could expect such strong regional signals for thermosteric variability also. Comparing the available proxy-based regional sea-level reconstructions with our model sea level would be a great exercise to

understand the role of ocean dynamics in driving the regional sea-level changes over the CE as well as the potential biases caused by a spatially non-uniform proxy network, and this would be a follow-up of this study.

Our model-based estimates are broadly consistent with the proxy-based GMSL reconstruction from earlier studies, despite a few disagreements combined with large uncertainties in the first millennium. For example, model results suggest that the

GMSL does not vary more than ±0.1 m during the PCE, a result consistent with the proxy-based sea-level reconstruction from Kopp et al., (2016), Kemp et al., (2018) and Walker et al., (2021, 2022). Our results also suggest that the GMSL generally rose over 1 – 500 CE, and there has been a long-term decline during 1000 – 1800 CE, an evolution consistent with the long-term global-mean surface temperature cooling in the PCE (Fig. 4b), as noted earlier by Kemp et al. (2011) and Kopp et al. (2016). However, as seen in our model, a pause in the generally rising GMSL around 600 CE is inconsistent with the proxy-based

GMSL reconstructions (see Figs. 4a, b). This inconsistency can be traced to the centennial-scale sea-level drop associated with thermosteric and Greenland contributions over the same period, as seen in Figure 2. Though the global-mean and Greenland surface temperature variability seems to control it (Figs. 2a&2c), it is difficult to make definitive conclusions due to the large uncertainty during the first millennium. Our results also quantify the current state of uncertainties in the individual contributions, which are particularly large for barystatic components over the first millennium. Challenges in incorporating

the impact of the ocean on marine ice sheets (both because of incomplete knowledge of ocean changes and ice sheet dynamical response to those changes) are serious limitations and a potentially major source of uncertainty. Additionally, the challenges in initializing the model with the right climate conditions at the beginning of the CE and reconciling the inherent uncertainties in the model input fields remain to be addressed in detail.


**Code and Data availability:** All new model outputs discussed in this paper are available on request from the corresponding author or directly at https://doi.org/10.5281/zenodo.7082320.

**Author contribution:** GN and HG[1] designed the study. GN analyzed the data and prepared the manuscript. HG[3] performed ice-sheet simulations. Glacier simulations were performed by DP and FM. All authors contributed to the discussions and writing of the manuscript.


**Acknowledgements.** This work was supported by Fonds National de la Recherche Scientifique (F.R.S.-FNRS-Belgium) in the framework of the project "Evaluating simulated centennial climate variability over the past millennium using global glacier modelling" (grant agreement PDR T.0028.18). Hugues Goosse is Research Director within the F.R.S.-FNRS. We acknowledge the World Climate Research Programme's Working Group on Coupled Modelling, which is responsible for CMIP, and we thank the climate modelling groups for producing and making available their model output. For CMIP, the US Department of Energy's Program for Climate Model Diagnosis and Intercomparison provides coordinating support and led the development of software infrastructure in partnership with the Global Organization for Earth System Science Portals. We thank Dr Jennifer Walker for providing reconstructed sea-level data on request. Computational resources have been provided by the supercomputing facilities of the Université catholique de Louvain (CISM/UCL) and the Consortium des Equipements de Calcul Intensif en Fédération Wallonie Bruxelles (CECI) funded by the Fond de la Recherche Scientique de Belgique (F.R.S.-FNRS) under convention 2.5020.11. Ice sheet simulations were performed on resources provided by UNINETT Sigma2 - the National Infrastructure for High Performance Computing and Data Storage in Norway through projects NN8006K, NN9560K, NS5011K, NS8006K and NS9560K. Heiko Goelzer acknowledges support from the Research Council of Norway through projects 270061, 295046 and 324639. FM acknowledges support from the Austrian Science Fund (FWF) grant P30256. We thank Christopher Piecuch and the two anonymous reviewers for providing critical comments and suggestions during the revision of this paper.

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

| Experiment | Thermosteric | Antarctic | Greenland | Glacier |
|---|---|---|---|---|
| **EXP-I** (1-2000 CE) | LOVECLIM (Goosse et al., 2010) [**10** members] ($T_o$, $S$) | IMAUICE (de Boer et al., 2014) | IMAUICE | OGGM (Maussion et al., 2019) |
| | | Stenni '17 [**1**] LMR [**1**] LOVECLIM [**1**] ($T_s$) | Kobashi '11 [**1**] LMR [**1**] LOVECLIM [**1**] ($T_s$) | LMR, CRU [**1**] ($T_s$, $P$) |
| **EXP-II** (850-2000 CE) | PMIP3/CMIP5 [**7**] CESM1 [**1**] ($T_o$, $S$) | IMAUICE | IMAUICE | OGGM |
| | | PMIP3/CMIP5 [**7**] LOVECLIM [**1**] CESM1 [**1**] ($T_o$, $T_s$) | PMIP3/CMIP5 [**7**] LOVECLIM [**1**] CESM1 [**1**] ($T_s$) | LMR, ERA5 [**1**] ($T_s$, $P$) |

**Table 1: Details of the two groups of model experiments presented in this study. The columns are split into two rows for the barystatic**
**components for EXP-I and EXP-II, highlighting the physical model used (top row) and the input used to run the model (bottom row). The number in square bracket shows the number of independent simulations made either using different models (e.g. PMIP model simulations) or based on the input field given to the physical model (ice-sheet/glacier model run with different input fields) for each component. Abbreviations (for the variables used) are: $T_o$ - Ocean Temperature, $S$ – Ocean salinity, $T_s$ – Surface Temperature, $P$ – Precipitation, Stenni '17 - Stenni et al., 2017, LMR – Last Millennium Reanalysis, Tardif et al., 2019, Kobashi '11**
**- Kobashi et al., 2011, CRU - Climatic Research Unit gridded Time Series v-4.01 (Harris et al., 2020), PMIP3/CMIP5 – Paleoclimate/Coupled Model Intercomparison Project (Taylor et al., 2012; Braconnot et al., 2012), CESM1 – Community Earth System Model (Otto-Bliesner et al., 2016), ERA5 - ECMWF atmospheric reanalysis - v5 (Hersbach et al., 2020).**

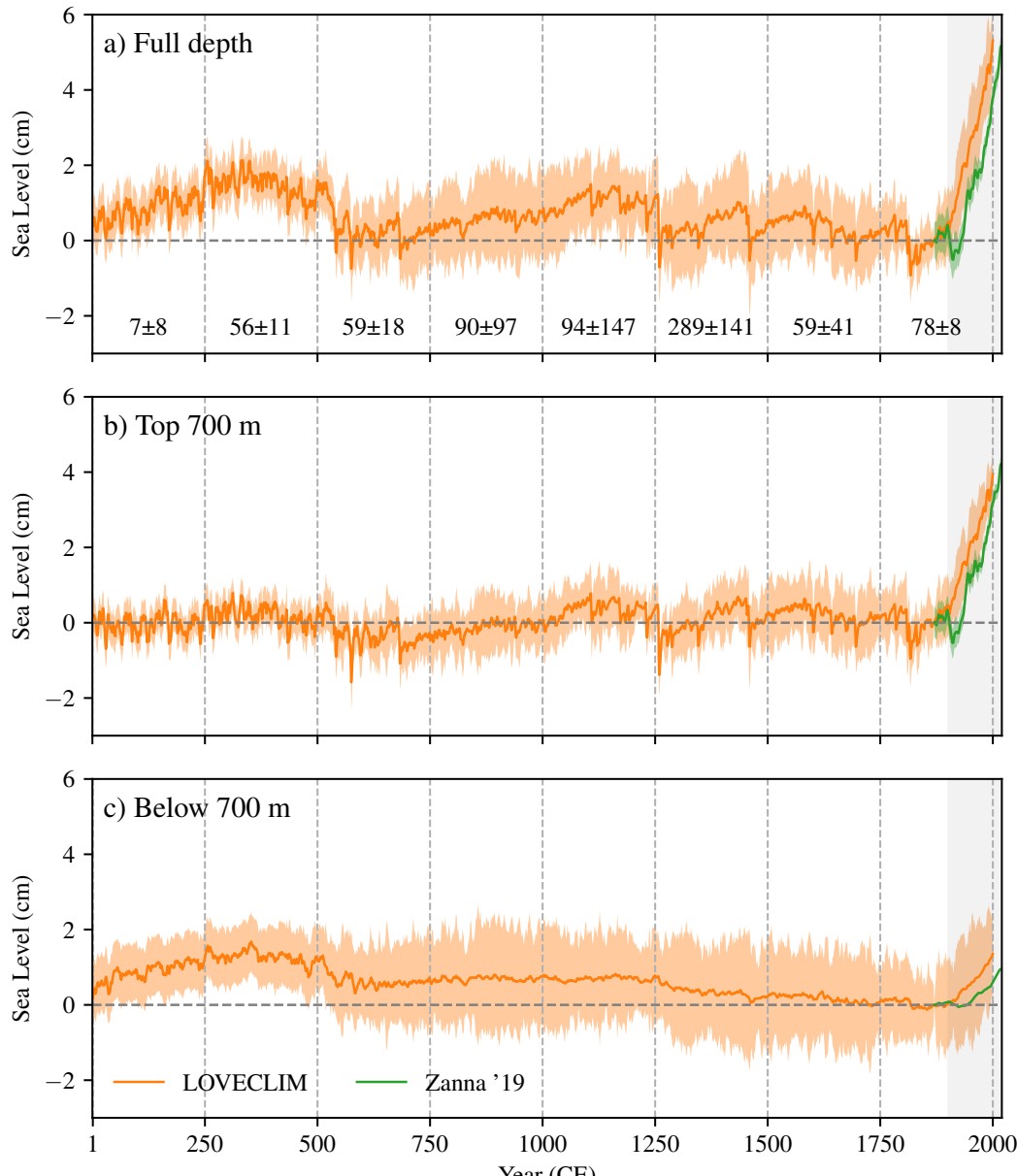

**Figure 1: Global-mean thermosteric sea level from LOVECLIM climate model simulations (orange; 1-2000 CE) and Zanna et al. (2019) reconstruction (green; 1870 - 2018) for a) full depth (b) top 700 m and c) below 700 m. All curves are referenced to 1870 CE (when the reconstruction begins), and the shading indicates the 1-$\sigma$ confidence level of the ensemble-mean curve. The contribution of change in the top 700 meters thermosteric level to that of the full depth (as percentage), estimated over chunks of 250-year periods (vertical dashed lines), is shown in panel a. The estimated 1900 – 2000 CE (period shown by grey shading) contribution of the upper 700 m for Zanna et al. (LOVECLIM) is 86% (74±8%).**

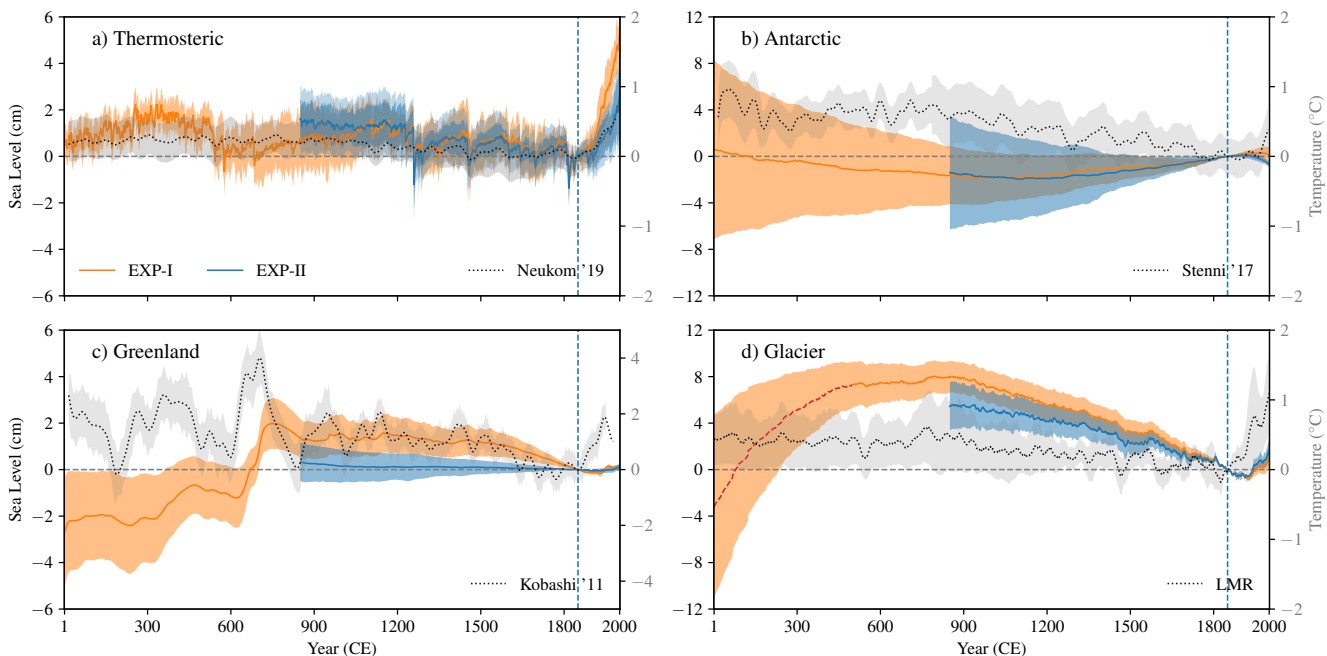

**Figure 2: Contributing components to GMSL. a) thermosteric, b) Antarctic ice sheet mass changes, c) Greenland ice sheet mass changes, and d) Glacier mass changes. Orange (EXP-I) and blue (EXP-II) curves represent the two sets of model simulations used in this study (see table 1). Global-mean surface temperature from Neukom et al. (2019) (a), surface temperature over Antarctica (b; from Stenni et al., 2017), Greenland (c; from Kobashi et al., 2011) and the glacier-area weighted surface temperature over 18 glaciated regions listed in the RGI (d; from LMR) are shown as dotted grey lines. The shading around sea-level curves, Greenland- and glacier-surface temperature indicates 1-$\sigma$ confidence level of the mean, and the shading of global-mean surface temperature (Antarctic surface temperature) shows 95% confidence level (2RMSE). Light blue shading of EXP-II thermosteric sea level (panel a) indicates the additional source of uncertainty arising from temperature changes below 700 meters (see text and Figure 1). The glacier contribution during 1 – 500 CE (EXP-I) is shown with a red-dotted line to indicate the uncertainty related to model initialization and spin-up (see section 2.3). All the curves are anomalies to 1841-1860 mean (blue dashed line at 1850 CE).**

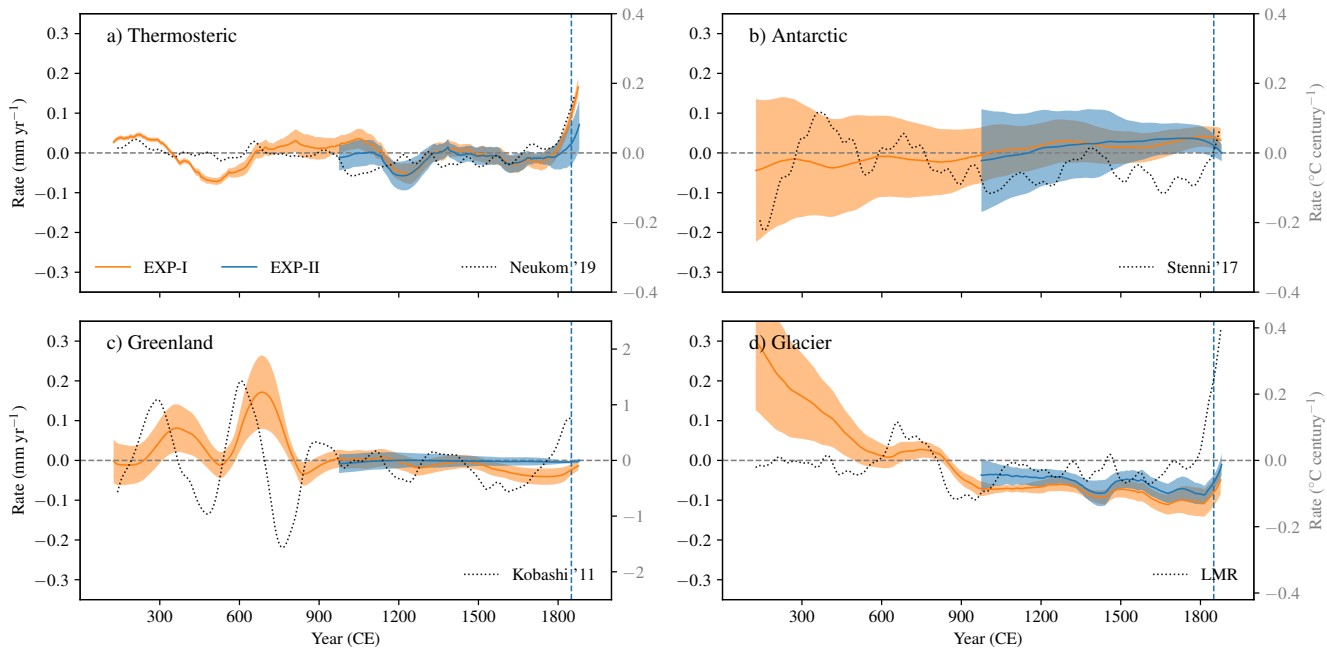

**Figure 3: 250-year moving rate of sea level for each of the components shown in Figure 2. a) thermosteric, b) Antarctic, c) Greenland and d) Glaciers mass balance. 250-year rate of global (a) and regional surface temperature (shown in figure 2) is also shown (dotted grey lines). The shading around sea-level rates indicates 1-$\sigma$ confidence level of the ensemble mean rate.**

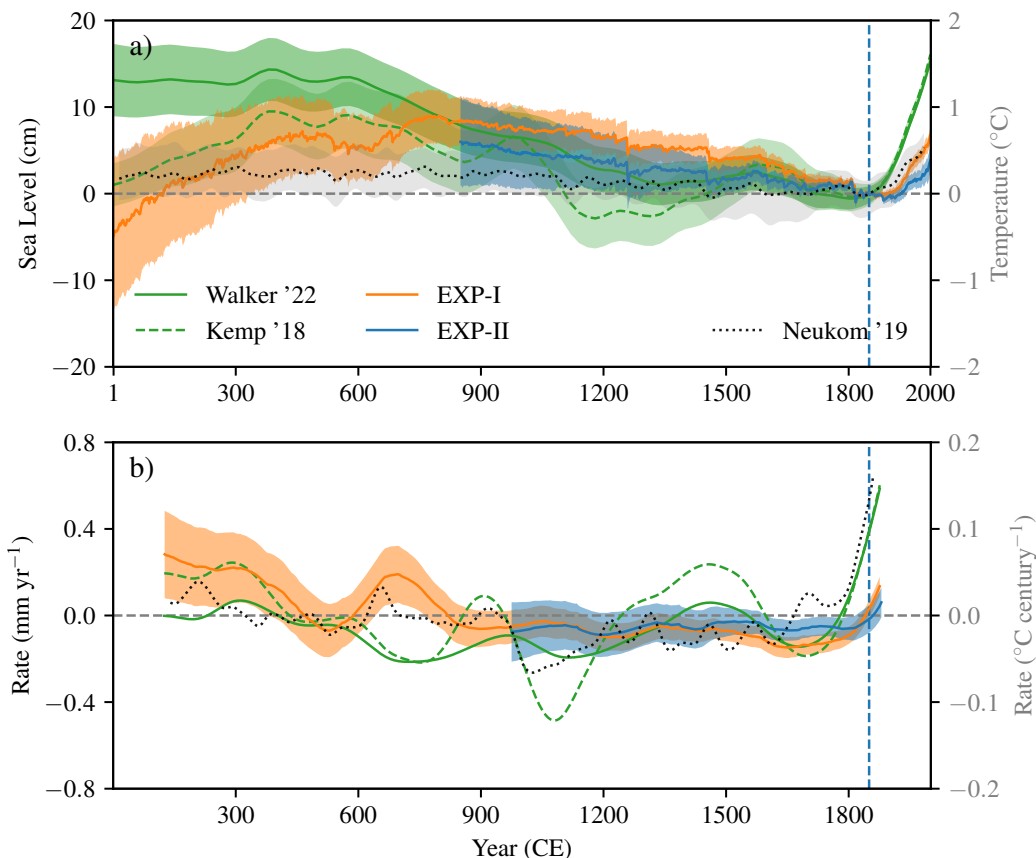

**Figure 4: a) Ensemble-mean GMSL estimated as the sum of the contributing processes from EXP-I (orange) and EXP-II (blue).**
**Proxy-based GMSL reconstruction from Walker et al. (2022; solid green), Kemp et al. (2018; dashed green), and the global-mean surface temperature from Neukom et al. (2019; dotted grey) are also shown. b) 250-year moving rate of GMSL and global-mean surface temperature curves shown in (a). Shading in a) and b) indicates 1-$\sigma$ confidence level of the ensemble mean curve except for global-mean surface temperature (a), for which the 95% confidence level is shown.**

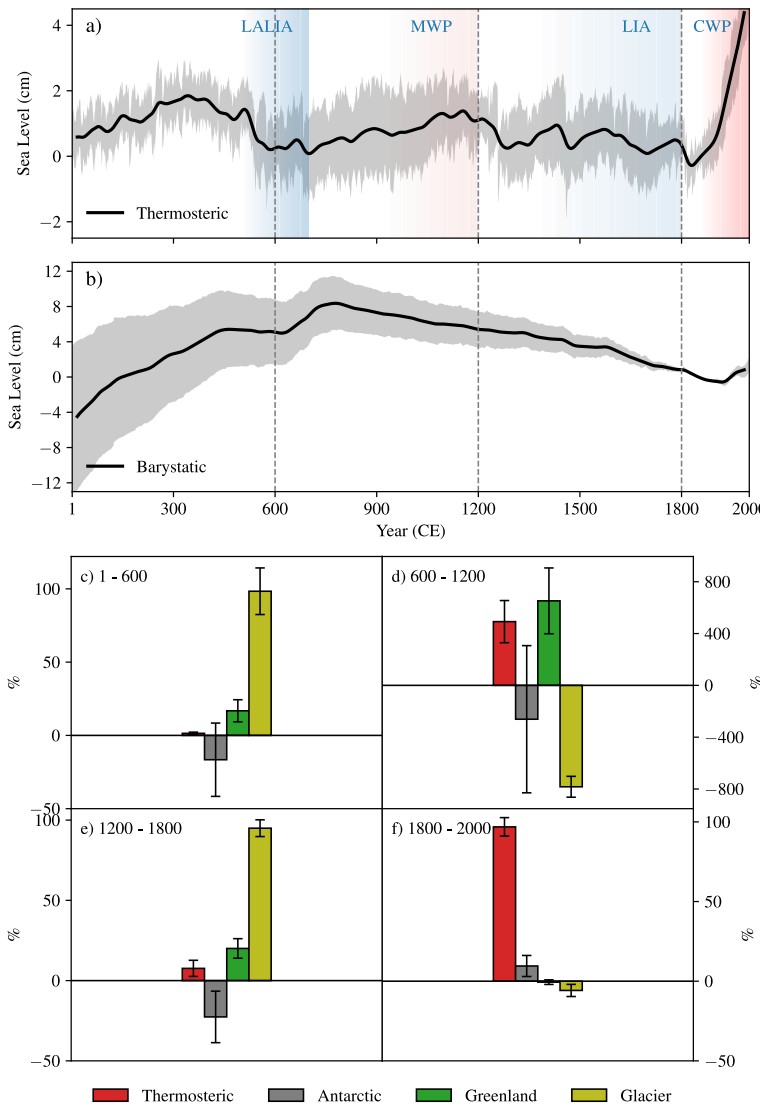

**Figure 5: Global-mean thermosteric (a) and barystatic (i.e. the sum of Antarctic, Greenland and glacier contribution; b) sea level with their 1-sigma uncertainty level from EXP-I (a 31-year smoothing is applied on the sea-level curves to focus on centennial time scales). We defined four time periods in the CE (1 − 600, 600 − 1200, 1200 − 1800 and 1800 − 2000; shown by dashed vertical lines in a, b) based on centennial-scale shifts seen in the global-mean thermosteric sea level. Those periods also mark the major sub-millennial climate epochs reported in the CE (The Late Antique Little Ice Age: LALIA (~ 600-700), The medieval warm period: MWP (~ 900 − 1300), The little ice age: LIA (~ 1300 − 1800) and the current warming period: CWP (post-1800). The respective contribution of thermosteric, ice-sheet and glacier mass-balance changes to model GMSL is estimated for these four periods and shown in panels c (1 − 600), d (600 − 1200), e (1200 − 1800) and f (1800 − 2000). Percentage contribution is calculated by a linear regression method, and the error bar represents the 1-$\sigma$ standard deviation of the contribution across the large ensemble.**



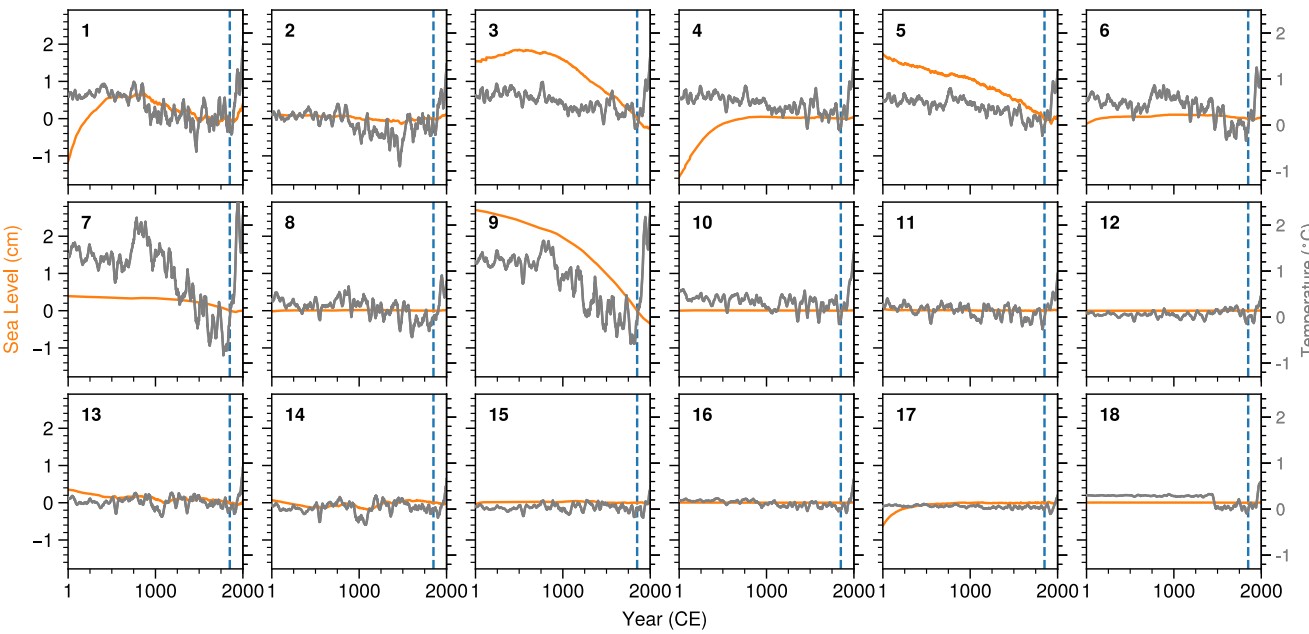


**Figure 6: Sea-level equivalent of glacier volume changes in the CE for the 18 RGI regions considered in the glacier model (orange) and the glacier-area weighted surface temperature (LMR) over each of the glacier regions in the RGI (grey). The number on the top left of each panel indicates the corresponding RGI region. Note that a 31-year low-pass filter is applied on surface temperature, but the original yearly simulation is shown for the glacier sea-level contribution. All curves are referenced to the 1841-1860 CE mean**

**(dashed blue line). Glacier regions, as listed in RGI, are 1. Alaska, 2. Western Canada and United States, 3. Arctic Canada North, 4. Arctic Canada South, 5. Greenland periphery, 6. Iceland, 7. Svalbard, 8. Scandinavia, 9. Russian Arctic, 10. North Asia, 11. Central Europe, 12. Caucasus and Middle East, 13. Central Asia, 14. South Asia West, 15. South Asia East, 16. Low latitudes, 17. Southern Andes, 18. New Zealand.**