# Peer review of "Process-based Estimate of Global-mean Sea-level Changes in the Common Era"

_Earth System Dynamics, 2022_

## Author Comment (AC1)

**We thank the Editor and the three Reviewers for critical comments and suggestions, which helped to revise this manuscript constructively. We provide our answers to each question below.**

**Reviewer query:** black

**Answer:** Blue

**Added/revised text to the main manuscript:** Blue, *italics*

/////////////////////////////////////////////////////////////////////////////////////////////////////////////////////////////

**Reviewer 1**

**General Comments**

This paper quantifies the contribution to Common Era global mean sea level (GMSL) from changes in ocean volume caused by temperature (salinity is not evaluated with justification provided) and from changes in ocean mass (Antarctica, Greenland, and glaciers are separated; land-water storage is not evaluated with justification provided). Each of these contributions is estimated, with uncertainty, through modeling. The sum of the components (GMSL) is reconciled with two, closely-related (and previously published) estimates of GMSL generated through application of a spatio-temporal model to proxy reconstructions. The similarity between modeled and reconstructed GMSL is compelling, except for a notable underestimate for GMSL change since ~1800 CE.

As the authors note, previous efforts to connect Common Era GMSL changes with other parts of the climate system (e.g., ice mass) largely focused on correlations (or lack of) with other proxy data. The application of process-based models in this paper therefore represents a welcome scientific advance and a substantive contribution toward our understanding of how and why GMSL changed during the Common Era. I recommend that it be published in Earth System Dynamics and hope that the open review forum will attract input from others to strengthen the paper further.

I am familiar with the proxy sea-level reconstructions that modeled GMSL is compared to and therefore my review focuses on that aspect of the paper. I am wholly unfamiliar with process-based models and cannot provide an expert evaluation of choices made within and among the models used.

We thank the reviewer for recommending the paper.

**Specific Comments**

1. Section 2.4 provides a short summary of the proxy-based GMSL reconstructions. I think that this section would benefit from a modest expansion to include some missing (but potentially important) information and some material that appears elsewhere in the paper already.

The Kopp (2016), Kemp (2018), and Walker (2021) GMSL reconstructions are largely iterations of a spatio-temporal statistical model applied to a growing database of Common Era proxy reconstructions. The authors might emphasize a little more that the GMSL reconstructions are less different models and more an evolution in the underlying data. Notably the GSML reconstruction became smoother over these sequential publications. The authors may also want to highlight that the geographic distribution of proxy records is very uneven, but that Kopp (2016) performed sensitivity tests to explore this influence. It is also important to recognize that GMSL is not a quantity that was reconstructed from a proxy, but is rather one component of the relative sea-level signal that is estimated during the record decomposition performed by the spatio-temporal model.

We thank the Reviewer for this suggestion. Iterating the comments from Reviewer 2 on the same, we have expanded the description of proxy-based sea-level reconstruction (section 2.4) as shown below. We also provide reference of Walker et al. (2022) as it is the latest update of the proxy sea-level database:

*GMSL derived from proxy-based sea-level reconstruction for the common era from Kopp et al. (2016), Kemp et al. (2018) and Walker et al. (2022) are considered for comparison with our model GMSL. Those GMSL reconstructions are iterations of a spatio-temporal statistical model applied to a growing database of Common Era proxy reconstructions. In this spatio-temporal model framework, GMSL is an estimate of global sea level obtained from the signal "common" to all of the sea-level records in the Common Era proxy database. Since the GMSL is the "globally uniform" term among sites from the spatio-temporal model, the method could give a true estimate of "GMSL" in the presence of spatially complete data. Consequently, the quality of the estimate depends on the geographic distribution of proxy records which is very uneven (however, some sensitivity tests to explore the effect of geographic distribution of proxy records has been done in Kopp et al. 2016). As the Walker et al. (2022) reconstruction is based on the latest update of the proxy sea-level database, and the Kemp et al. (2018) and Kopp et al. (2016) curves do not differ much over the CE, we show GMSL from Walker et al. (2022) and Kemp et al. (2018) in our model comparison. Also note that in Kemp et al. (2018), the GMSL during -100 – 100 CE is made equal to GMSL over 1600 – 1800 CE to avoid a spurious regional sea-level trend component. However, such a constraint is not employed in Walker et al. (2022) reconstruction. As a result, there is an apparent difference between the GMSL curves in these two reconstructions before ~ 600 CE.*

Text on lines 363-365 and 374-375 could be moved into section 2.4.

*Those texts are moved as suggested.*

In this section it might also be appropriate to highlight when notable differences exist between the two GMSL reconstructions (e.g., before ~600 CE).

*We agree. As mentioned above, we added this information in section 2.4 itself as:*

*Also note that in Kemp et al. (2018), the GMSL during -100 – 100 CE is made equal to GMSL over 1600 – 1800 CE to avoid a spurious regional sea-level trend component. However, such a constraint is not employed in Walker et al. (2022) reconstruction. As a result, there is an apparent difference between the GMSL curves in these two reconstructions before ~ 600 CE.*

2. Sea-level fingerprints. Changing the mass of water stored on land as ice results a in fingerprint of sea level change. Although it's beyond the paper's focus on GMSL, I think it would be interesting and helpful to show the fingerprints (from individual sources and their sum) that occur as a consequence of the modeled changes in mass from Greenland, Antarctica, and the 18 regions of glaciers. The fingerprints could be compared to the distribution of proxy records, or to estimates of regional sea level trends. It is possible that fingerprinting could help inform model choice if proxy records support/refute particular melt histories. As minimum could the regional contributions from glaciers be provided as a supplemental output for others to convert into sea-level fingerprints.

*We thank the Reviewer for this suggestion. However, as mentioned in the comment this is beyond the scope of this first paper and we kept the regional patterns of absolute (thermosteric and barystatic) and relative (GRD and GIA fingerprints) sea-level change during the Common Era as a perspective of this paper. We agree that comparing the available proxy sea-level reconstructions with our model sea level would be a great exercise to understand the role of ocean dynamics in driving the regional sea-level changes over the Common Era as well as the potential biases caused by a spatially non-uniform proxy network. We restrained ourselves showing spatial changes in this paper since those exercises requires additional analyses and discussion and deserve for us a specific paper. We totally understand the importance of characterizing the land vertical movements and corresponding sea-level fingerprints due to last millennium mass redistribution, and we will consider it in a following paper. Nevertheless, all the model simulation we have will be available on a public domain and on personal communication once the paper is published. We have added a few sentences in the conclusion section of the paper to convey this perspective.*

3. Glaciers appear to be the single most important driver of Common Era GMSL change, but also the most problematic to model and quantify. Please could the authors show the contributions from the 18

different regions of glaciers. In Figure 3D the glacier contribution is shown against global temperature, which the paper does acknowledge (line 460) is an imperfect comparison since glaciers respond to regional climate. Could the glacier contribution from the 18 regions be compared to regional climate from Neukom et al (2019)?

Sea-level contribution from 18 regions defined in the Randolph Glacier Inventory (RGI; Pfeffer et al., 2014) along with the surface temperature (ST) averaged over each RGI region from Last Millennium Reanalysis (LMR v2; Tardif et al., 2019) data are shown below. We use LMR as it is used to drive the glacier model (section 2.3) and is more consistent; on the other hand, it is hard to obtain regionally downscaled ST from Neukom et al. (2019). We have included this figure (S1; shown below) as a supplementary figure in the paper.

[Figure]

*S1: Sea-level contribution (orange) from 18 glacier regions defined in the RGI (excluding Antarctic/sub-Antarctic) and the glacier-area weighted surface temperature (blue) over each region extracted from LMR-v2. All the curves are anomalies with respect to 1841 − 1860 mean. Note that a 31-year running mean is applied on surface temperature and the original annual mean is shown for the glacier sea-level contribution.*

The modeled glacier contribution is large (even described as "remarkable" on line 456), which presumably indicates that some proportion of the 18 regions were behaving in a temporally-coherent fashion (growing/melting at the same time as one another), or possibly that a subset of regions dominates the glacier signal. Simultaneous contributions across multiple regions in the pre-anthropogenic Common Era might be surprising since a principal conclusion of Neukom et al. (2019) is that temperature trends were not spatially-coherent during this time. I was therefore surprised to see such a large and sustained glacier contribution, because the Neukom et al (2019) analysis led me to think that as one region warmed, another cooled and therefore that the change in glacier mass (and its contribution to GMSL) would be moderated. This is even more surprising because Neukom et al. (2019) conclude that twentieth-century warming is the only temperature signal which is globally coherent and therefore would affect all the glacier regions simultaneously, yet the contribution to GMSL from glaciers is smaller and slower that it was during times of incoherent temperature variability. I think some regional analysis of glaciers by region would be a useful addition to this paper.

We thank the Reviewer for this comment. As shown in S1 (which is included as a supplementary figure to the paper), there is considerable regional variability in the history of both glaciers and surface temperature throughout the Common Era. The global sum of glacier contribution is indeed the result of very different regional signals. Linking surface temperature with regional glacier changes would be difficult without further diagnoses, nevertheless, we assume that the changes of surface temperature and glaciers might occur over distinct time scales. Neukom et al. (2019) focused on the absence of consistent warm and cold phases at multidecadal to centennial timescales. For instance, the surface temperature shows strong decadal to multi-decadal variability both regionally (S1) and globally. On the other hand, the large-scale glacier changes in the CE are mostly a centennial to multi-centennial response, for which the spatial consistency might appear higher (S1 and Fig. 2d). Bringing discussion on regional changes is probably out of the scope of this present paper which is focused on globally averaged signals (in a similar way, we restrain from describing the regional contribution of thermal expansion in different oceanic basins). However, the salient features of S1 would be added in the discussion part. Also, all the time series shown in S1 will be made available for anyone who wish to explore the regional glacier variability further, once the paper is accepted for publication. We also would like to mention that the global-mean surface temperature is replaced by glacier-area weighted surface temperature averaged over the 18 regions from LMR, in Fig. 2d.

The authors note that modeled GMSL is considerably less than observed and reconstructed twentieth-century rise. The difference is attributed to underestimating the barystatic contribution, especially from glaciers. In particular, the distribution and size of glaciers at the start of the Common Era was set (by necessity) to be the same as that observed in ~2000 CE, despite anthropogenic warming having already impacted them significantly by ~2000 CE (including some glaciers being lost – line 425 – and therefore

missing from the modeled contribution throughout the Common Era presumably). The authors discuss how this effect modeled GMSL since ~1800 CE, but offer less insight into how the problem could bias GMSL estimates before 1800 CE (other than suggesting that the very large contribution from glaciers before ~400 CE could be a spin-up effect from using ~2000 CE as the initial state). I would be interested to read an expanded discussion about how modeled GMSL appears to be an underestimate for the past 200 years, but agrees well with reconstructed GMSL at least for ~800-1800 CE despite the difficulties with glaciers. For example, the difference between GMSL as modeled and reconstructed by Walker is large before ~600 CE. Could (and how) might glaciers solve/cause this discrepancy? If some glaciers are missing, does this mean that the modeled contribution from glaciers is a minimum, and would somehow adding them back in to the GMSL calculation fix the discrepancy since ~1800 CE at the expense of creating a new discrepancy before ~1800 CE?

We thank the reviewer for his insightful comments. There are large uncertainties and limitations in simulating the glacier changes for the Common Era and we have noted some of them in the methods and discussion parts (Line 423; as the reviewer pointed out). Given those large uncertainties (especially the initial glacier distribution and right climate forcing at the beginning of the CE), we emphasize that it is virtually impossible to have a quantitative validation, especially during the PCE. We totally agree with the reviewer that the prescribed initial volume may have a huge impact on the rest of the glacier evolution, however, initializing the model with a new state as an alternative can also bring uncertainty (including the spin up to such a new initial state). We have added a few sentences more in the discussion to highlight these uncertainties.

**Technical Corrections**

Line 49: The Walker et al. paper is cited as a 2020 publication, but it is listed (correctly) as a 2021 publication in the reference list.

This is corrected in the text.

Line 217: Title needs a capital letter.

Corrected

The 20$^{th}$ century is variously referred to as "20thC" (section 4.1), "twentieth century" (e.g., line 30), or "20$^{th}$ century" (e.g., line 204). These could me made consistent throughout the manuscript.

We made it consistent by using *twentieth century*

Line 196: "R" should be changed to "r" for consistency with other titles.

Corrected

I found Figure 1 to be a little confusing. Readers might find it easier if a third panel was added to show the "below 700m" component rather than including it in panel B which is described initially in the caption as the "top 700m". Or alternatively place the below 700 m, above 700m and total in a single panel.

We agree. This figure is redrawn as suggested.

The use of two y-axes in figure 5a to show the same quantity (sea level, cm) at different scales made the figure difficult to use.

We agree. The main objective with figure 5a is to show the consistent changes in global-mean thermosteric sea level with those climate epochs (shown by light shading in the panel) and to quantify the respective contribution of individual processes during each epoch (bar plots). The net barystatic curve shown in figure 5a does not bring anything new so that we discarded it from the panel.

---

## Author Comment (AC2)

**We thank the Editor and the three Reviewers for critical comments and suggestions, which helped to revise this manuscript constructively. We provide our answers to each question below.**

**Reviewer query:** black

**Answer:** Blue

**Added/revised text to the main manuscript:** Blue, *italics*

/////////////////////////////////////////////////////////////////////////////////////////////////////////////////////////////////

**Reviewer 2#**

**General comments**

This manuscript produces a new analysis of global-mean sea level change over the common era using process-based modeling with an examination of thermosteric and barystatic (Antarctic, Greenland, and glaciers) contributions through time. The authors compare their modeled GMSL with proxy reconstructions of global sea level and find general agreement, although the model-based estimate underestimates twentieth-century GMSL. They find that glaciers acted as the dominant source of GMSL changes during the common era; however, the uncertainties were large especially in the last millennium. The paper is generally clear and well written and while there are some large uncertainties in the results, it is valuable to have new process model-based estimates of GMSL to compare with proxy reconstructions and to further understand the relative contributions of processes driving GMSL changes over longer timescales through the common era.

I would recommend the manuscript to be published in Earth System Dynamics if the following several points could be addressed to improve the discussion of the results and comparison with proxy reconstructions. My comments focus on these aspects of the paper, as I cannot expertly comment on the intricacies of the process modeling methods themselves.

We thank the reviewer for this comment.

**Specific comments**

The last paragraph of the introduction mainly refers to analysis during the PCE (except for Ln 71 which says "changes over the CE") which is inconsistent. However, the results and discussion do cover the entire CE, not just the PCE, so I would suggest altering the text accordingly.

Thanks. We made a change and refer to CE in the text.

Because the authors clearly state questions in the introduction that the paper will attempt to answer (Ln 71-73), I would expect clearer answers to each of these questions in the discussion or at the conclusion of the paper. Especially concerning the major sources of uncertainty – while the large uncertainties are referenced throughout the paper, it would be helpful to clearly state the sources of these uncertainties at the conclusion of the paper and suggestions for how to minimize them in future work.

Thanks for this suggestion. We have added a final section (conclusion) in the paper to summarize the main conclusions of the paper and provide a general discussion of the uncertainties as pointed by the Reviewer.

Section 2.4 could be strengthened to explain the proxy-based reconstructions of global sea level – such as the proxy data that was used, the basic methods with spatiotemporal modeling. Specific details like Ln 361-365 describing the different curves could be moved to section 2.4 instead. It would also be helpful to more completely explain the Kopp/Kemp/Walker global reconstruction – that it is an estimate of global sea level via the signal common to all of the sea-level records in the Common Era proxy database. It is therefore the "globally uniform" term among sites from the spatiotemporal model, and not exactly an estimate of GMSL. The Kopp/Kemp/Walker method could give a true estimate of "GMSL" in the presence of spatially complete data.

Thanks! Also considering comments from Reviewer 1 on the same, we have expanded the description of proxy sea-level reconstructions in section 2.4 as shown below:

*GMSL derived from proxy-based sea-level reconstruction for the common era from Kopp et al. (2016), Kemp et al. (2018) and Walker et al. (2022) are considered for comparison with our model GMSL. Those GMSL reconstructions are iterations of a spatio-temporal statistical model applied to a growing database of Common Era proxy reconstructions. In this spatio-temporal model framework, GMSL is an estimate of global sea level obtained from the signal "common" to all of the sea-level records in the Common Era proxy database. Since the GMSL is the "globally uniform" term among sites from the spatio-temporal model, the method could give a true estimate of "GMSL" in the presence of spatially*

*complete data. Consequently, the quality of the estimate depends on the geographic distribution of proxy records which is very uneven (however, some sensitivity tests to explore the effect of geographic distribution of proxy records has been done in Kopp et al. 2016). As the Walker et al. (2022) reconstruction is based on the latest update of the proxy sea-level database, and the Kemp et al. (2018) and Kopp et al. (2016) curves do not differ much over the CE, we show GMSL from Walker et al. (2022) and Kemp et al. (2018) in our model comparison. Also note that in Kemp et al. (2018), the GMSL during -100 – 100 CE is made equal to GMSL over 1600 – 1800 CE to avoid a spurious regional sea-level trend component. However, such a constraint is not employed in Walker et al. (2022) reconstruction. As a result, there is an apparent difference between the GMSL curves in these two reconstructions before ~ 600 CE.*

The descriptions of the proxy-based reconstructions of global sea level need to be corrected. In Ln 361-365 describing the methodological constraint, it is correct that Kemp et al. (2018) used this constraint. However, Walker et al. (2021) also utilized this constraint so this needs to be corrected in Ln 365. The constraint was used for all of the analysis in Walker et al. (2021) and the global curve shown in that paper uses the constraint. A supplemental figure in Walker et al. (2021) shows the global curve without using the constraint – which is the curve that is shown in this paper in comparison to the process model estimate. This needs to be made clear throughout this manuscript and in Figure 4. Alternatively, Walker et al. (2022) could be referenced, which did remove the constraint for the analysis and so the global sea-level results do not include the constraint – this would be the equivalent global curve to what is actually shown in this paper.

Walker, J.S., Kopp, R.E., Little, C.M. et al. Timing of emergence of modern rates of sea- level rise by 1863. Nat Commun 13, 966 (2022). https://doi.org/10.1038/s41467-022-28564-6

We thank the reviewer for pointing this out. To avoid any confusion and to reduce text, we refer to Walker et al. (2022) while discussing figure 4.

In Ln 339-342, could the authors speculate as to what would cause the differing response of the Greenland and Antarctic ice sheets to surface temperature changes? Or provide any references that also support these findings?

Greenland surface temperature and its sea-level contribution shows an in-phase variability. Higher temperatures induce more melting of the Greenland ice sheet and thus a sea level rise (Lines 334 – 340)). The relationship of Antarctic sea level contribution and surface temperature, on the other hand, was described as 'inverse' as the temperature increase over Antarctica leads to increased mass accumulation and a decrease in sea level (line 330 – 333). *"The surface temperature over Antarctica in the past two millennia (Stenni et al., 2017) exhibits an inverse relationship to sea level over multi-centennial periods*

*(Fig. 2b). Our experimental design can explain this relationship as a warmer climate generally enhances precipitation over Antarctica and decreases the GMSL (Frieler et al., 2015; Medley and Thomas, 2019)*". The dominance of different processes explains the differing response of the two ice sheets and we have shifted Lines 330 – 333 to after Line 342 to make this difference between the two ice sheets easier to follow.

In Ln 360 (and throughout the paper) I think it would be more clear and helpful to refer to "reconstructions" as "proxy-based reconstructions" instead.

We have modified the paper as suggested by the Reviewer.

In Ln 405-409, first a positive contribution is related to GMSL rise in Ln 405, meaning a negative contribution is related to GMSL fall in Ln 406. So how is in Ln 408-409 "All the GMSL components except Antarctic ice sheet have a positive contribution to net GMSL fall during 1200-1800 CE" supposed to be interpreted?

What is shown in panels 5 b-e is the ratio of the rate of individual contribution to total GMSL rate (in terms of percentage). Hence, a positive contribution simply means that the rate sign of both the component and GMSL are same (i.e. both rates are either positive or negative). And, a negative contribution means that ratio is negative (GMSL and component rate have different signs). Hence, "*All the GMSL components except Antarctic ice sheet have a positive contribution to net GMSL fall during 1200-1800 CE*" means that the net GMSL and sea level from individual components (except Antarctic) is falling (ratio is positive) during the 1200-1800 CE.

I understand the uncertainties and limitations using the process-based model, but I find it difficult to put too much weight on the results for the PCE, when the 20th century global sea-level estimates are inconsistent with reconstructions and observations and are underestimated to a degree that there is not even overlap within the uncertainties. If the model was altered/improved to match the observations/reconstructions in the twentieth-century, how would this change GMSL and the relative contributions of driving processes (especially glaciers) over the rest of the PCE? Can a more formal list of improvements be recommended to address this discrepancy? How much of this is due to the initial conditions in the model and is there a way that these could be adjusted? I think these questions need to be addressed more completely in the discussion.

The underestimation of twentieth-century model GMSL comes from the barystatic components (see for instance the agreement between model and reconstructed twentieth-century thermosteric sea level in fig. 1). We have uncertainties on model initialization, reference climate state used, and forcing fields in the

common era. Alternative strategies are possible as the reviewer suggests. The models state in 1800 results from the simulation since 1CE and thus may integrates biases over this all period, in particular due to model drift and uncertainties in the forcing. We could 'correct' the state in 1800 to have better results over the last 2 centuries but, for example, the glacier distribution around ~1800 to initialize the model is not well-known and of the new model drift it will induce at the start of the simulation is hard to estimate. Such experiments would be interesting but our goal is to provide a consistent set up over the full millennium, with uncertainties clearly highlighted, not to have the most realistic set up for the twentieth-century (which is the aim of other existing studies (e.g. Marzeion et al. 2015; Frederikse et al. 2020) as we have noted in section 4.1: "*Uncertainties on ice sheet simulations are even larger and what we present here is a qualitative description of ice sheet changes in the common era based on Physics but it's quantitative assessments (for example the twentieth-century change) require further improvements (better constraining the climate forcing, developing paleo data etc.)*". We have added a few more sentences to highlight these aspects in the discussion part.

**Technical corrections**

Ln 49, 50, 361, 365: these should reference Walker et al. 2021, not 2020

It is corrected.

Ln 424: 'focused' spelled incorrectly

Thanks. It is corrected.

Figure 1: the caption says the Zanna et al., (2019) reconstruction is blue, but it is green on the figure

Thanks. It is corrected.

Figure 4b: would be helpful to show the uncertainties in the rates for the Kemp/Walker/Neukom curves

The confidence level of the model GMSL rate is estimated using the large ensemble members (section 2.5.2 L262). On the other hand, single confidence level (another single curve) is available for the reconstructions. Moving rate of this additional curve would not be an uncertainty estimate of the original rate curve. Also, showing the range of all the curves would make the figure hard to read. Given these, we restrain ourselves showing the range of rate curves for the reconstructions.

---

## Author Comment (AC3)

**We thank the Editor and the three Reviewers for critical comments and suggestions, which helped to revise this manuscript constructively. We provide our answers to each question below.**

**Reviewer query:** black

**Answer:** Blue

**Added/revised text to the main manuscript:** Blue, *italics*

/////////////////////////////////////////////////////////////////////////////////////////////////////////////////////////////////////

**Reviewer 3#**

A review of "Process-based Estimate of Global-mean Sea-level Changes in the Common Era" by Nidheesh, Goosse, Parkes, Goelzer, Maussion, and Marzeion

The authors use models to quantify thermosteric effects from ocean heat storage and barystatic effects from land ice changes on long-term global-mean sea-level (GMSL) fluctuations during the preindustrial Common Era (PCE). They compare their results to proxy reconstructions of PCE GMSL changes from Kopp, Kemp, and Walker. One of the authors' main conclusions is that glaciers made dominant contributions to GMSL changes during the PCE.

I'm a sea-level scientist with training in physical oceanography. I don't have expertise in modeling land ice. Thus, I restrict my review mainly to sections on thermosteric effects, and recommend the Editor solicits reviews from experts in ice-sheet and glacier modeling.

I really liked this study. Something like it has been needed for a while. The past decade has seen real advances in the community's ability to quantify PCE GMSL changes from proxy reconstructions (from the likes of Kemp, Kopp, Walker, and others). But there's been a total lack of modeling studies to complement those observational studies. This paper starts to fill that gap: it won't be the last word on the subject, but it takes the logical first steps, and therefore deserves to be published after some minor revisions.

We thank the reviewer for recommending the study.

**Specific comments**

References to Walker et al. (2020) should be to Walker et al. (2021)

*Thanks. It has been corrected.*

Section 2.1. The authors describe how they estimate global-mean thermosteric sea level from PMIP3/CMIP5 temperature and salinity. Why not just use the zostoga global-mean thermosteric sea-level diagnostic variable made available by several PMIP3/CMIP5 groups? Also, the authors should identify precisely which PMIP3/CMIP5 model simulations they use (they only identify GISS-ES-R; were other models used?).

*"ZOSTOGA" available for most of the PMIP3 models exhibits strong climate drift and we do not have a corresponding control run ZOSTOGA available for PMIP models to correct the drift. That is the prime reason we estimated the GMTSL using T, S simulation and restricted the computation to the upper 700 meters (we find that the drift is mostly associated with deep-layer temperature adjustments). In addition to that, understanding regional steric variations is also a perspective of this research and estimating thermosteric and halosteric fields from gridded temperature and salinity is required. Please note that the list of PMIP3/CMIP5 models used in this work is given in the supporting information which is mentioned in the main text (section 2.1; line 89).*

The authors consider LOVECLIM, PMIP/CMIP models, and the reconstruction of Zanna et al. (2019). Is that all of the relevant data sources for ocean warming and thermosteric effects during the Common Era? Are there other ocean reconstructions that could also be brought in to corroborate the story they're telling?

*Thanks for this question. This point has been clarified in the method section. Hence, we expanded section 2.4 as follows:*

*We also compare our model GMTSL with the reconstructed GMTSL estimates from Zanna et al. (2019) over 1870 – 2018. Since Zanna et al. (2019) already compared their reconstruction to different observation-based oceanic heat content estimates (e.g. Levitus et al. 2012; Ishii et al. 2017), we do not show all those available products in this paper for the twentieth century comparison. Reconstructions of ocean temperatures over the CE are limited to either sea surface temperature derived from paleoceanography (proxy) data (e.g. PAGES Ocean2k Synthesis Data; McGregor et al. 2015) or spatially averaged oceanic heat content estimates generated through inverse modelling and using available instrumental and paleo-data (Gebbie and Huybers 2019). Though such datasets have been shown useful to understand certain key features of ocean climate variability during the Common Era, they do not provide a direct estimate of the contribution of ocean changes to GMSL. Hence, we do not*

*attempt to compare our model thermosteric variability with any of those datasets in this paper. Also, as the GMSL reconstruction from Walker et al. (2022) and Kemp et al. (2018) already incorporated the tide-gauge-based twentieth-century GMSL, we do not show those available twentieth-century GMSL reconstructions in this paper.*

Section 2.5.2 Uncertainty on rest of the processes. I find this whole section unclear, ad hoc, and arbitrary. Can the authors please explain more the basic rationale and provide references for their methods when possible? In particular, I'm confused what their uncertainty quantification is supposed to represent. What missing process do they imagine they're accounting for by adding the autocorrelated noise, for example?

As we state in the beginning of section 2.5.2, because of the limited number of independent estimates of GMSL from each contributing process, it is hard to quantify (and show) the uncertainty in a consistent manner. Hence, the basic rationale of generating 1000 synthetic curves from existing / available curves by perturbing them using white noise is to have a consistent set of sea-level time series for each of the contributing processes considered. We have estimated the glacier uncertainty in an independent way (but still by producing thousand members; section 2.5.1). This kind of perturbation is not supposed to bring any additional specific process that misses in our modelling experiments but simply to acknowledge the remaining uncertainty (e.g. uncertainty arises from model initialization, inputs or differences in model physics) using a simple framework and to propagate the overall uncertainty to the final GMSL curve in a consistent way. One specific case, is the addition of the RMSE of thermosteric variability below 700 meters (derived from LOVECLIM) to the perturbation standard deviation of PMIP3 thermosteric curves (upper 700 meters) to specifically account for the uncertainty associated with missing variability below 700 meters (as noted in Line 246). We also agree that the entire procedure of uncertainty estimation might be a bit arbitrary, however, some choices have to be made and a similar approach could be seen in other studies (e.g. Frederikse et al. 2020).

Section 3.1

Line 266ff. Can the authors speculate on the high-frequency (decadal) global-mean thermosteric variability apparent in the LOVECLIM solution that is unrelated to volcanism? Is it related to ENSO or another global mode of natural climate variation (e.g., Hamlington et al., 2020, PNAS)?

We have not addressed specifically this point in our analysis which is focused on longer term changes. Our first guess is that such internal variability such as the one associated with ENSO (whose amplitude is very small in LOVECLIM) should be damped in the *ensemble mean* that we show in the paper. Those decadal signals could also be a response to other forcing, for instance, the small volcanos which are not discussed in the manuscript. We prefer to restrain ourselves commenting more on them without proper diagnosis.

Lines 283ff. Can the authors speculate on the mechanisms of these changes and when or why upper-ocean and deep-ocean effects may be opposing or reinforcing? More generally, some discussion of the physics involved, rather than just a tabulation of numbers, would be informative.

Thanks for this suggestion. We added the following sentences (Lines 283ff) to inform this:

*The lag in the lower-layer thermosteric rise compared to recent warming of the upper ocean (~ since 1850 CE; Fig. 1b) could be due to the extending deep-layer cooling from LIA, as shown in Gebbie and Huybers (2019). Similarly, a rise in the upper 700 m thermosteric sea level during 1250 – 1400 CE might be a rebound of the upper ocean from volcanic cooling (strong 1257 eruption), but the deep ocean has still cooled during this period (Fig. 1b). In general, the differing response of thermosteric response in the upper and lower ocean indicates two distinct time scales of ocean response, the deep layer being much slower than the upper ocean.*

Line 290. What is the plus/minus values?

As noted in section 2.5.2, we compute the rate, standard deviation and budget of GMSL *for each ensemble member* and subsequently derive the mean and confidence intervals from the large ensemble. Hence, ± indicate the 1-std of the mean standard deviation across members.

Line 302 and elsewhere. Why the italics on Roman Warm Period? Also on the next line it should be Antiquity not Antique.

Thanks. It is corrected.

Line 305ff. Is LOVECLIM model output available to say more about what drove these multi-centennial changes? Were they the long-term effects of volcanism? Changes in insolation? Again, some physical insights would be useful.

The main forcing in LOVECLIM is volcanic and the insolation has a weak impact on oceanic variability in annual mean (see for instance the supplementary document of McGregor et al. 2015).

Line 373ff. The more muted variability in the Walker et al. (2021) results relative to the Kemp et al. (2018) result may be owing to different prior assumptions made in the two studies with regard to dominant timescales of variability (i.e., what time smoothing is implied by the respective versions of the empirical spatiotemporal model).

As we note in the main text, there is an apparent difference between the GMSL curves of Kemp et al. (2018) and Walker et al. (2021) before ~ 600 CE, which may manifest as a unique centennial-scale variability in either of the reconstruction prior to ~ 600 CE. This difference comes because in Kemp et al. (2018), the GMSL during -100 – 100 CE is made equal to GMSL over 1600 – 1800 CE to avoid a spurious regional sea-level trend (arise from GIA) component. However, such a constraint is not employed in Walker et al. (2022) reconstruction. In addition to that, Walker et al. (2021) curve is smoother (less variability) compared to other reconstructions as the reviewer pointed out, and this could primarily due to the fact that Walker et al. (2021) employed the latest updated database of sea-level reconstructions in the model. The regional sea-level signal is expected to possess more lower-frequency variability and including data from more locations could potentially cancel out the variability in the global mean. We would like to restrain ourselves going deeper into features of sea-level reconstruction and focus on our model global-mean signal.

Line 393ff. Have the authors identified why their model results with respect to global-mean thermosteric versus barystatic contributions during the twentieth century differ so greatly from estimates of Slangen et al. (2017; Surveys in Geophysics) and Frederikse et al. (2020; Nature)? The authors point out the differences several times, but it would be good to know why these differences exist, and whether they bear on the confidence we have in their simulations of the PCE.

The underestimation of twentieth-century model GMSL comes from the barystatic components (see for instance the agreement between model and reconstructed twentieth-century thermosteric sea level in fig. 1). We have uncertainties on model initialization, reference climate state used, and forcing fields in the common era. If we consider glacier simulations, for instance, the models state in 1800 results from the simulation since 1CE and thus may integrates biases over this all period, in particular due to model drift and uncertainties in the forcing. We could 'correct' the state in 1800 to have better results over the last 2 centuries but, for example, the glacier distribution around ~1800 to initialize the model is not well-known and of the new model drift it will induce at the start of the simulation is hard to estimate. Such experiments would be interesting but our goal is to provide a consistent set up over the full millennium, with uncertainties clearly highlighted, not to have the most realistic set up for the twentieth-century (which is the aim of other existing studies (e.g. Marzeion et al. 2015; Frederikse et al. 2020). As we have noted in section 4.1: "*Uncertainties on ice sheet simulations are even larger and what we present here is a qualitative description of ice sheet changes in the common era based on Physics but it's quantitative assessments (for example the twentieth-century change) require further improvements (better constraining the climate forcing, developing paleo data etc.)*". We have added a few more sentences to highlight these aspects in the discussion part.

---

## Author Response (AR2)

**Reviewer query:** black

**Answer:** Blue

**Added/revised text to the main manuscript:** Blue, *italics*

**Reviewer 1**

**General Comments**

My review focuses on the changes to the manuscript made in response to my original review. The authors have largely addressed my queries through edits made to the manuscript and rebuttals in their response letter. I recommend publication of this article following some minor changes.

We thank the reviewer for his earlier comments and suggestions.

Specific Points

Section 2.4: Original review comment adequately addressed.

Thanks

Calculation of sea-level fingerprints: Adequately addressed in response letter.

Thanks

Glaciers: The authors produced a new figure to show the breakdown of the contribution to global mean sea level from glaciers in 18 different regions (with the 19th region being excluded from analysis with justification provided). The new presentation of results is included as supplemental figure 3 and it is exactly the type of result I had hoped to see. However:

(1) I encourage the authors to promote this important figure to the main manuscript given the key role of glaciers (if the number of figures is limited, figures 2 and 3 could be combined to make space).

We thank the reviewer for this suggestion and his earlier comments on glacier contribution. We added supplementary figure S3 to the main text (now Figure 6) as the reviewer suggested.

(2) I suspect some readers may not be familiar with the regions by number in the Randolph Glacier Inventory. Please include a map panel, or at least supplement the panel legends with a name to go with the number.

We agree. Glacier region names corresponding to the RGI numbers are given in the caption of figure 6.

(3) Please could the authors standardize the y-axes among panels (the sea level axes currently vary by more than two orders of magnitude). The key result shown in the new figure is that the main contribution to pre-industrial Common Era global mean sea-level change is not glaciers generally, but rather glaciers in a handful of regions (e.g., regions 3, 5, and 9; while regions 10-12 effectively make no contribution). Standardizing the temperature axes would also give a more intuitive sense of how glacier mass balance and temperature correlations are variable/consistent among regions. Use of LMR temperature appears reasonable.

Thanks. The new figure 6 uses a standardized scale to highlight these points. We also added a few lines in the discussion to convey that the glacier contribution is not uniform across regions but largely comes from a handful of regions, as:

*As evident from Fig. 6, not all glacier regions contributed equally to the GMSL, but a few areas (e.g. Greenland periphery, Russian Arctic) dominate the others (e.g. North Asia, Low latitudes). The distribution of glacier sea-level contribution in the CE seems to relate to the glacier initial volume distribution (which is the twentieth-century glacier volume distribution as given in Farinotti et al., 2019).*

The revised discussion text (e.g., line 520) about spatial and temporal variability in the glacier contribution is very helpful and addresses the points raised in my review (or at least notes some of the queries, even if answers are not easy to come by).

Thanks. In fact, the points raised by the reviewer opened the discussion more widely and improved the quality of the discussion.

Technical Corrections

Each of the technical corrections I originally proposed were addressed by the authors. Below I list some very minor editorial points that I noticed.

Line 79: A comma following "terrestrial water storage" would be helpful.

Corrected

Line 192: Should read "we therefore use ~2000 CE".

Corrected

**Reviewer 2#**

**General Comments**

I appreciate the authors' efforts in addressing my previous comments and find that the manuscript has been greatly improved through the revision process and therefore recommend it for publication. Below are a number of minor, mostly technical, comments that I would suggest for the final version of the manuscript.

We thank the reviewer for his earlier comments and suggestions.

Ln 220 add 'the' in "not employed in 'the' Walker et al…."

Corrected

Ln 220-221 and Ln 421-422 While the difference between the two reconstruction curves is likely due in part to the removal of the constraint, it should also be noted in the text that the database was updated with additional proxy records from the Kemp curve to the Walker curve, so that could also be having an influence on the difference between the two curves.

Thanks. We added a sentence to indicate this in the text (ln 216)

Ln 233 add reference to tide gauge-based 20th century GMSL

Hay, C. C., Morrow, E., Kopp, R. E. & Mitrovica, J. X. Probabilistic reanalysis of twentieth-century sea-level rise. Nature 517, 481–484 (2015).

Thanks. We provide reference for Hay et al. (2015) to indicate the twentieth-century sea-level reconstruction.

Ln 277-279 this wording doesn't seem quite right if I'm reading it correctly… would suggest rewording to "Note that this method does not include any additional specific processes that are missing in our modelling experiments but acknowledges the remaining uncertainty (e.g. uncertainty arising from model initialization, different input data, or differences in model physics)"

Changed as suggested.

Ln 330 could you add a reference about this 1257 eruption? Or any more details?

Thanks. It is the strongest eruption in the last millennium (Mt Samalas in Indonesia). We have provided the reference for this in the text.

Sigl, M., Winstrup, M., McConnell, J. et al.: Timing and climate forcing of volcanic eruptions for the past 2,500 years, Nature, 523, 543–549, 2015

Ln 425 add 'the' in "However, 'the' Walker et al…"

Okay

Ln 453 should 'contributed by' be 'attributed to'?

Yes, corrected.

Ln 529 "contribution from Antarctica" rather than "contribution of Antarctic"

Corrected.

Ln 588 would suggest rewording to describe what "it" is in this sentence

We rewrote this sentence as, "*Those centennial-scale changes during the PCE indicates that the twentieth-century GMSL rise may also include a response to such natural variations*"

Ln 591 do not need to say 'sources' twice

Corrected

Ln 607 should reference Walker et al 2022, not 2021

Walker et al. (2022) is also included

**Reviewer 3#**

**General Comments**

A second review of "Process-based Estimate of Global-mean Sea-level Changes in the Common Era" by Gangadharan and coauthors.

I thank the authors for addressing my earlier comments. This study should be published after very minor revisions. I don't need to see another iteration. I'd like to congratulate the authors on a nice study and a valuable contribution to the community.

Kind regards,

Christopher Piecuch, Woods Hole Oceanographic Institution

We thank Christopher Piecuch for his valuable comments during the revision. We do acknowledge him in the paper along with the two anonymous reviewers as:

*"We thank Christopher Piecuch and the two anonymous reviewers for providing critical comments and suggestions during the revision of this paper"*

Minor points

1. On line 214, the authors say that Kemp et al. (2018) impose a constraint of no net GMSL change between ~0 and ~1700 CE to avoid a spurious regional trend. This isn't exactly true. Those authors (as well as Kopp et al. 2016) impose that constraint to avoid large uncertainties on the global trend, not regional trends.

Thanks. This sentence has been rewritten as: "Also, in Kemp et al. (2018), the GMSL during -100 – 100 CE is made equal to GMSL over 1600 – 1800 CE to avoid a spurious global sea-level trend component originating from regional changes.

2. More generally, I suggest all of the authors to give a careful proof reading of the manuscript for grammar and syntax. I haven't taken the time to enumerate all such instances, but the authors should ensure there aren't any remaining typos or awkward phrasings.

Sure